# Amyloid aggregates accumulate in melanoma metastasis modulating YAP activity

Vittoria Matafora[1] iD, Francesco Farris[1] iD, Umberto Restuccia[1,†], Simone Tamburri[1,‡], Giuseppe Martano[1], Clara Bernardelli[1,§], Andrea Sofia[1,2], Federica Pisati[1,3], Francesca Casagrande[1], Luca Lazzari[1], Silvia Marsoni[1], Emanuela Bonoldi[4] & Angela Bachi[1,*] iD

## Abstract

Melanoma progression is generally associated with increased transcriptional activity mediated by the Yes-associated protein (YAP). Mechanical signals from the extracellular matrix are sensed by YAP, which then activates the expression of proliferative genes, promoting melanoma progression and drug resistance. Which extracellular signals induce mechanotransduction, and how this is mediated, is not completely understood. Here, using secretome analyses, we reveal the extracellular accumulation of amyloidogenic proteins, i.e. premelanosome protein (PMEL), in metastatic melanoma, together with proteins that assist amyloid maturation into fibrils. We also confirm the accumulation of amyloid-like aggregates, similar to those detected in Alzheimer disease, in metastatic cell lines, as well as in human melanoma biopsies. Mechanistically, beta-secretase 2 (BACE2) regulates the maturation of these aggregates, which in turn induce YAP activity. We also demonstrate that recombinant PMEL fibrils are sufficient to induce mechanotransduction, triggering YAP signaling. Finally, we demonstrate that BACE inhibition affects cell proliferation and increases drug sensitivity, highlighting the importance of amyloids for melanoma survival, and the use of beta-secretase inhibitors as potential therapeutic approach for metastatic melanoma.

Keywords amyloid; BACE2; mechanosignaling; metastasis; secretome
Subject Categories Cancer; Molecular Biology of Disease; Signal Transduction

## Introduction

Melanoma is the most aggressive cutaneous cancer, resulting from the transformation and proliferation of skin melanocytes (Shain &

Bastian, 2016). While melanoma only accounts for 1% of skin cancers, it is responsible for the majority of skin cancer deaths with an incidence in rapid increase over the past 30 years. In Europe, melanoma accounts for about 150,000 new cases and 27,147 deaths/year (Globocan 2018, https://gco.iarc.fr/today/fact-sheets-populations). Approximately 85% of melanomas are diagnosed at early stages when the tumor is thin and surgery is curative in > 95% of cases (American Cancer Society, www.cancer.org). For advanced disease, either unresectable or already metastatic, the therapeutic landscape has benefitted since few years from an unprecedented number of new drugs (e.g. immune checkpoint inhibitors and small-molecule targeted drugs) which have significantly improved the prognosis of advanced patient which is otherwise dismal. That notwithstanding, < 30% of these cases reaches the 5-year landmark disease free, clearly indicating that a deeper insight into the biology of melanomas is an unmet clinical need (Guy et al, 2015; Pasquali et al, 2018). The main cause of death is widespread metastases, which commonly develop in regional lymph nodes or in distant organs. Melanoma cells travel along external vessel lattices by regulating adhesion molecules, matrix metalloproteases, chemokines, and growth factors. After steadied in the metastatic sites, melanoma cells develop mechanisms that protect them against the attack of the immune system (Zbytek et al, 2008).

Progression to metastatic melanoma is accompanied by increased cell stiffness and the acquisition of higher plasticity by tumor cells, due to their ability to control stiffness in response to diverse adhesion conditions (Weder et al, 2014). During melanoma development, tumor cells are exposed to various types of extracellular matrix (ECM) such as tenascin-C, fibronectin (Frey et al, 2011), and collagen fibers which led to an overall more rigid tumor microenvironment (Yu et al, 2011). Stiffness was suggested to control phenotypic states and to contribute to the acquisition of a malignant phenotype. Indeed, in epithelial cancers, an extracellular environment characterized by softer matrix enables

1 IFOM- FIRC Institute of Molecular Oncology, Milan, Italy
2 University of Insubria, Varese, Italy
3 Cogentech SRL Benefit Corporation, Milan, Italy
4 Department of Laboratory Medicine, Division of Pathology, Grande Ospedale Metropolitano Niguarda, Milan, Italy
 *Corresponding author. Tel: +39 02574303873; E-mail: angela.bachi@ifom.eu
 †Present address: ADIENNE Pharma & Biotech, Caponago, Italy
 ‡Present address: Department of Experimental Oncology, IEO-European Institute of Oncology IRCCS, Milan, Italy
 §Present address: Fondazione Politecnico di Milano, Italy

differentiation, while a stiffer matrix increases proliferation (Lee *et al*, 2017). Increased ECM rigidity might also serve as "safe haven" for melanoma cells, protecting them from the effects of chemotherapy; as such, these drug-induced biomechanical niches foster tumor growth and residual disease favoring melanoma resistance (Hirata *et al*, 2015). It was observed that BRAF inhibitors do not only act on tumor cells but also on the neighboring tumor fibroblasts, paradoxically activating them to produce a stiff, collagen-rich ECM. Melanoma cells fast respond to this new microenvironment by increasing ECM attachment, and reactivating MAPK signaling in a BRAF-independent manner. Understanding how cancer cell-derived ECM is regulated, and how it participates in tumor microenvironment remodeling and signaling is critical for developing novel cancer treatment strategies. In this context, secretome studies from tumor and stromal cells provide novel insights in the understanding of the cross-talk between cells within the tumor microenvironment, since they are very sensitive in revealing the key effectors required for the establishment of pre-metastatic niches (Kaplan *et al*, 2005; Karagiannis *et al*, 2010; Blanco *et al*, 2012).

We therefore sought to explore tumor melanoma microenvironment by secretome analysis, investigating the molecular mechanism behind malignant matrix stiffening.

## Results

### *In vitro* model of metastatic and primitive melanoma

With the aim to understand the functional pathways that differentiate tumor microenvironment of metastatic and primitive phenotype, we investigated two pairs of matched melanoma cell lines. In particular, IGR39 and IGR37 were derived from primitive tumor and lymph node metastasis, respectively, collected from the same 26-year-old male patient. Similarly, WM115 and WM266.4 matched cell lines were derived from cutaneous primitive tumor and skin metastasis, respectively, from the same 55-year-old female patient (Fig 1A). Despite their common origin, these cell lines display different phenotypes. In both cases, metastasis-derived cell lines showed a faster growth rate and increased ability to undergo rapid division compared to the matched primitive tumor-derived cell line (Figs 1B and EV1A). Differences in morphology were also denoted between the two matched cell lines: Metastasis-derived IGR37 appeared with a short, elongated shape with a spontaneous predisposition to form clusters, while primitive tumor-derived IGR39 remained commonly isolated, displaying higher number of branches and branch elongations (Fig 1C). On the other hand, IGR39 had higher mobility compared with IGR37 when monitored live using time-lapse microscopy (Fig 1D, Movie EV1 and EV2). All these data suggest that cells isolated from metastatic tumors grow faster, but move slower than primitive tumors, symptomatic of a proliferative phenotype (Hoek *et al*, 2008) for IGR37 and an invasive phenotype for IGR39.

### Global analysis of secreted proteins reveals specific signatures of tumor microenvironment

To investigate the molecular composition of melanoma secretome, we performed a global analysis of the proteins secreted by metastasis-derived (IGR37 and WM266.4) and primitive tumor-derived (IGR39 and WM115) cells lines. To differentiate between proteins that were secreted *versus* the ones present in the serum from cultured conditions, a triple SILAC was performed. We labeled the proteins coming from primitive and metastatic cell line, respectively, with medium and heavy amino acids (Fig 2A). For each sample, we analyzed the conditioned medium (CM) after 24 h of serum deprivation to avoid contamination of high abundant proteins as albumin. We checked for the absence of proteins derived from dead cells by measuring the viability of the cell lines upon starvation, and we confirmed that none of them suffered that condition (viability > 95%, Fig EV1B). As far as secreted proteins are highly glycosylated and this modification might mask proteolytic sites hampering protein digestion, we set up a novel method, named Secret3D (Secretome De-glycosylation Double Digestion protocol), where a de-glycosylation step (PNGase) was added prior to protein digestion performed with double proteolysis to increase protein coverage. Our method enables the unambiguous identification of secreted proteins with high efficiency and quantitative accuracy (Dataset EV1 and EV2). As reported in Fig 2A, by examining an equivalent of 500,000 cells, we identified 2,356 proteins in the SILAC IGR37/IGR39, and 2,157 proteins in the SILAC WM266.4/WM115, increasing four times the yield compared to digestion without de-glycosylation (Fig EV1C, Dataset EV3 and EV4, Peptide Atlas repository; glycosylated peptides represent about one-fourth of the entire dataset, Fig EV1D), and improving the proteome coverage if compared to existing methods (Liberato *et al*, 2018). All proteomics analyses were done in biological duplicate, and for each biological replicate two technical replicates were performed. By statistical analysis, 270 proteins were found to be differentially secreted in IGR39 versus IGR37. A parallel analysis was conducted in WM115 versus WM 266.4 cells where the number of differentially secreted protein was 185 (Fig 2B and C, Dataset EV1 and EV2). Results were clustered in order to highlight secretome commonalities. Despite showing only a partial overlap of the differentially secreted proteins (Fig 2D), both primitive tumor-derived cell lines specifically secrete proteins belonging to ECM matrix degradation (MMP1), Wnt signaling pathway (WNT5a), TGFB signaling pathway (TPM4), proteoglycan degradation (SPOCK1), and platelet activation (VEGFC, SERPINE1, EDIL3). These proteins are in agreement with their invasive phenotype, as primitive tumor cells are able to move and invade through the basement membrane or through the vessels walls (Fig 1D).

Conversely, metastasis-derived cell lines secrete specific proteins belonging to ECM deposition (LAMA1, LAMB2, SPP1), cell adhesion molecule (MCAM, CD276, EMILIN2), lipid transporters (APOE, APOD, PLTP, VLDLR), and melanogenesis-related proteins (DCT, KIT1, PMEL; Fig 2D and E). Interestingly, APOE is the most secreted proteins in both metastatic cell lines. APOE is a lipoprotein whose primary function is transporting cholesterol, but it also controls the formation of protein aggregates in Alzheimer disease (AD) through the regulation of amyloid-β (Aβ) metabolism, aggregation, and deposition. Together with APOE, in metastatic secretome, we found enriched SORT1 and QPCT, proteins known to have a role in

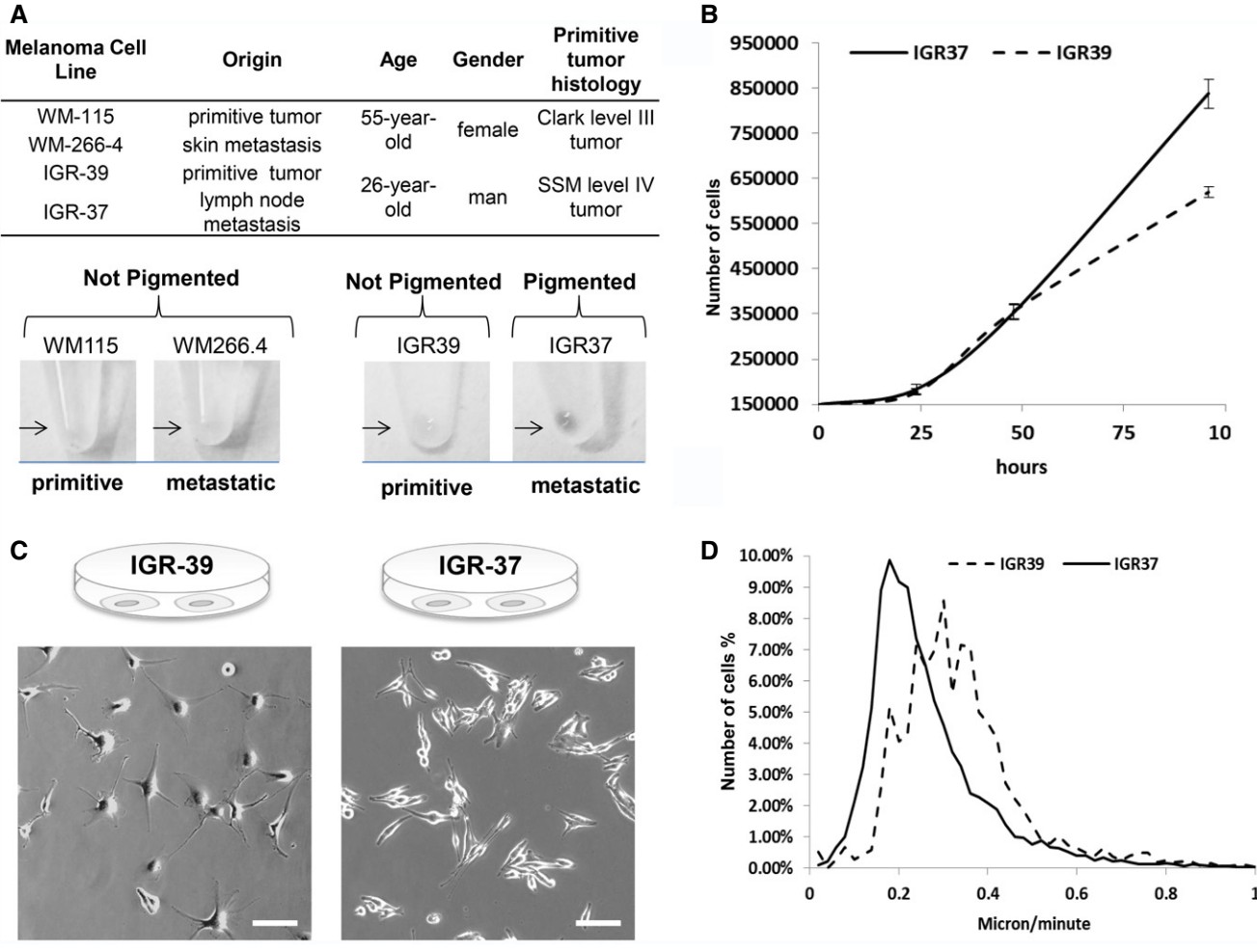

**Figure 1. Analysis of a cellular system for primitive and metastatic melanoma.**

A   Characteristics of IGR39/IGR37 and WM115/WM266.4 melanoma cell lines (Upper Panel). Pigmentation of melanoma cells pellets (lower panel).
B   Growth curve of primitive IGR39 and metastatic IGR37 cell line (*N* = 3 biological replicates). Data are mean ± SD.
C   Phase-contrast images of IGR39 and IGR37 cells. Scale bar is 100 μm.
D   Analysis of IGR39 and IGR37 speed of migration by time-lapse microscopy.

facilitating Aβ metabolism (Gunn *et al*, 2010; Morawski *et al*, 2014). Notably, in pigmented cells, PMEL maturation into fibrils is also mediated by APOE and shares similarities with amyloid-β maturation (Van Niel *et al*, 2015). PMEL amyloid fibrils are known to serve as scaffold for the polymerization of melanin within melanosomes. We found all the above proteins, PMEL included, enriched in the extracellular space highlighting the possibility of fibril formation extracellularly, i.e. plaques.

To test this hypothesis, we used a protein aggregation detection dye (Proteostat). Both pairs of IRGs and WMs matched cell lines were explored: Proteostat staining highlighted an enrichment of aggregated proteins in the metastasis- versus primitive tumor-derived cell lines both intracellularly and in the extracellular space (Figs 2F and G, EV1E and F). To note, amyloid fibrils, together with adhesive proteins, may contribute to the formation of a highly fibrotic extracellular environment specifically in the metastatic cell lines.

**Proliferative and invasive protein signatures are conserved across melanoma cell lines**

The secretome signature that distinguishes primitive versus metastatic melanoma was further validated on another cohort of cell lines derived from different patients (Fig EV2A). The secretion rate of all the cell lines analyzed (Fig EV2B) differs greatly from each other and inversely correlates with growth rate (Fig EV2C). Analysis was performed using a mixed model based on merged results from dataset normalized either by total number of cells or total protein content. We selected the proteins that were statistically significantly regulated with both approaches. Despite different melanoma cells have different doubling times, different shapes, and different protein secretion rates, we observed highly conserved metastatic and primitive signatures (Fig EV2D and E, Datsets EV5–EV8, Peptide Atlas repository). Among the differentially secreted proteins, five were conserved in all the primitive cell lines analyzed: MMP1, CBX5,

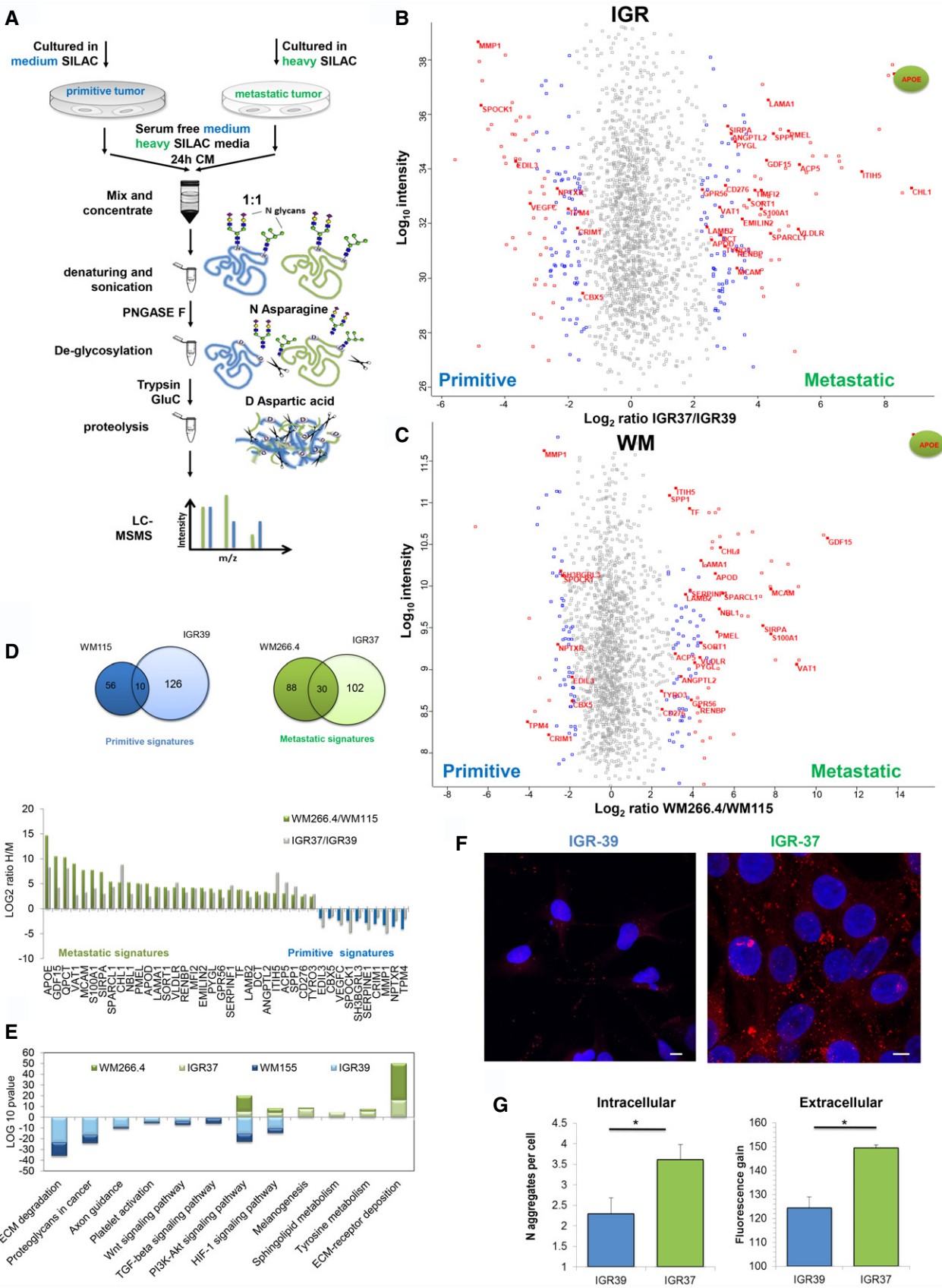

**Figure 2.**

**Figure 2. Proteomic analysis of the secretome from primitive and metastatic melanoma cells.**

A   MS workflow of Secret3D: Secretome De-glycosylation Double Digestion protocol.
B   Scatter plot of identified and quantified proteins in the secretome of primitive IGR39 and metastatic IGR37. Red dots represent proteins that were significant with FDR < 0.05, and blue dots represent proteins with P < 0.05.
C   Scatter plot of identified and quantified proteins in the secretome of primitive WM115 and metastatic WM266.4. Red dots represent proteins that were significant with FDR < 0.05, and blue dots represent proteins with P < 0.05.
D   (Left) Venn diagram of the significant proteins shared by both IGR and WM cell lines. (Right) Histograms representing the metastatic and primitive signatures H/M ratios.
E   KEGG pathway enrichment analysis of the significant proteins.
F   Confocal fluorescence images of Proteostat (1:2,000, red spots) and DAPI staining (blue), scale bar is 10 μm.
G   Quantitation of aggregates/cell in IGRs cell lines by immunofluorescence analysis, left panel; fluorescence gain of soluble proteins treated with Proteostat reagent, right panel. (T-test analysis, *P < 0.05, N = 3 biological replicates, data are mean ± SD).

NPTXR, CRIM1, and TPM4, and sixteen proteins were present in all the metastatic cell lines. In accordance with our previous data, APOE, PMEL, QPCT, and SORT1 were specifically secreted in all metastatic tumor cell lines together with proteins involved in ECM deposition and adhesive proteins supporting the hypothesis of amyloid fibril deposition (Fig EV2F).

By interrogating the same melanoma cell lines analyzed above in Broad-Novartis Cancer Cell Line Encyclopedia (CCLE), the proteins involved in amyloid fibrils maturation found enriched in the metastatic cell lines were found also enriched at the transcriptomic level (Fig EV3A). Together with the conserved secretome profile, we found other proteins involved in protein aggregation, such as APP, APLP2, and APLP1, secreted by melanoma in a cell line-specific manner. Formation of protein aggregates in the metastatic cell lines was visualized with Proteostat (Fig EV3B).

## APOE and PMEL proteins are overrepresented in the secretome of metastatic melanoma

As discussed before, APOE is the most abundant protein in metastatic secretome. APOE is known to be regulated by liverX receptor (LXR) which is activated by 24- and 25-hydroxycholesterol. In order to verify the involvement of APOE and of cholesterol metabolites in metastatic melanoma, we measured the level of oxysterols in melanoma cells. Indeed, both 24 and 25-hydroxycholesterol were more abundant in metastatic melanoma cells than in primitive tumor cells, thus possibly explaining the proteomic data in matched cell lines (Fig 3A). Importantly, these data were confirmed also in the other cohort of melanoma cells, where 24-hydroxycholesterol showed the best correlation with APOE levels (Fig 3B), indicating a specific cholesterol metabolism activation in metastatic melanoma. These evidences sustain the activation of LXR receptor in the regulation of APOE expression in metastatic melanoma, similarly to what reported in astrocytes by Abildayeva and coworkers (Abildayeva et al, 2006), thus enhancing the maturation of PMEL into amyloid fibrils.

We then verified the presence of PMEL amyloidogenic fragments in metastatic melanoma cells by Western blot analysis using an antibody that recognizes the mature form of the protein. As reported in Fig 3C, PMEL is exclusively expressed in metastatic cells and not in their primitive counterparts and the molecular weight corresponds to the mature form.

## Amyloid-like aggregated proteins accumulate in metastatic lesions of melanoma patients

Starting from the observation that amyloid-like aggregated proteins were found enriched particularly in the secretome of metastatic

melanoma cell lines, we explored if protein aggregates are present also in melanoma patients' tissues. To this aim, we examined samples deriving from primitive tumors and metastases. Archival formalin-fixed paraffin-embedded (FFPE) specimens collected from skin primitive tumors and differently localized metastatic sites (i.e. skin, brain, and lung) were stained with Proteostat and analyzed by high-resolution large-scale mosaic/confocal imaging. In both primitive and metastatic tumor samples, we detected a weak or absent signal of protein aggregates localized in the healthy region surrounding tumor tissue (Fig 4A). Conversely, in primitive tumors, protein aggregates were found spreading along the tissues in small isolated regions inside the tumor area (Fig 4B). Interestingly, a much higher representation of protein aggregates was detected in metastatic melanoma tissues, without any difference of metastases localization (lung, brain, and subcutaneous skin) (Fig 4C and D). These data support the hypothesis that progression from primitive to metastatic melanoma is accompanied by increased proteins aggregation.

In details, we observed that protein aggregates appear as dot-like structure on tumor tissue, clearly defining tumor edge (Fig 4E). Moreover, the number of protein aggregates, quantitated by counting the number of dots per cell, was significantly enriched in metastatic lesions compared to primitive tumors (Fig 4F). Interestingly, the presence of protein aggregates is not related to pigmentation, as there is no correlation between melanin (hematoxylin–eosin) and Proteostat staining (Fig EV3C); on the other hand, a remarkable correlation with cell proliferation (Ki-67 staining) can be observed (Fig EV3D), symptomatic of a more proliferative phenotype for the metastatic tissues (Hoek et al, 2008). These results are in accordance with the proliferative phenotype observed in metastatic versus primitive cell lines (Figs 1 and EV1A).

## BACE inhibition impairs the amyloidogenic machinery in metastatic melanoma cell lines

After demonstrating the enrichment of protein aggregates in metastatic tissues, we wondered if it would be possible to interfere with their production and affect metastasis behavior. The beta-secretase (BACE 1 and 2) enzymes are known to be involved in the formation of protein amyloids. Indeed, PMEL and APP are cleaved by BACE 2 and 1, respectively, and are able to form mature amyloid fibrils through an APOE-mediated process (Rochin et al, 2013). Notably, by interrogating gene expression profiling in TCGA and GTEx dataset, we found that BACE 2 is overexpressed in melanoma more than in any other cancer type (Fig EV3E) and correlates with a poor prognosis (Fig EV3F). Moreover, melanoma is also characterized by

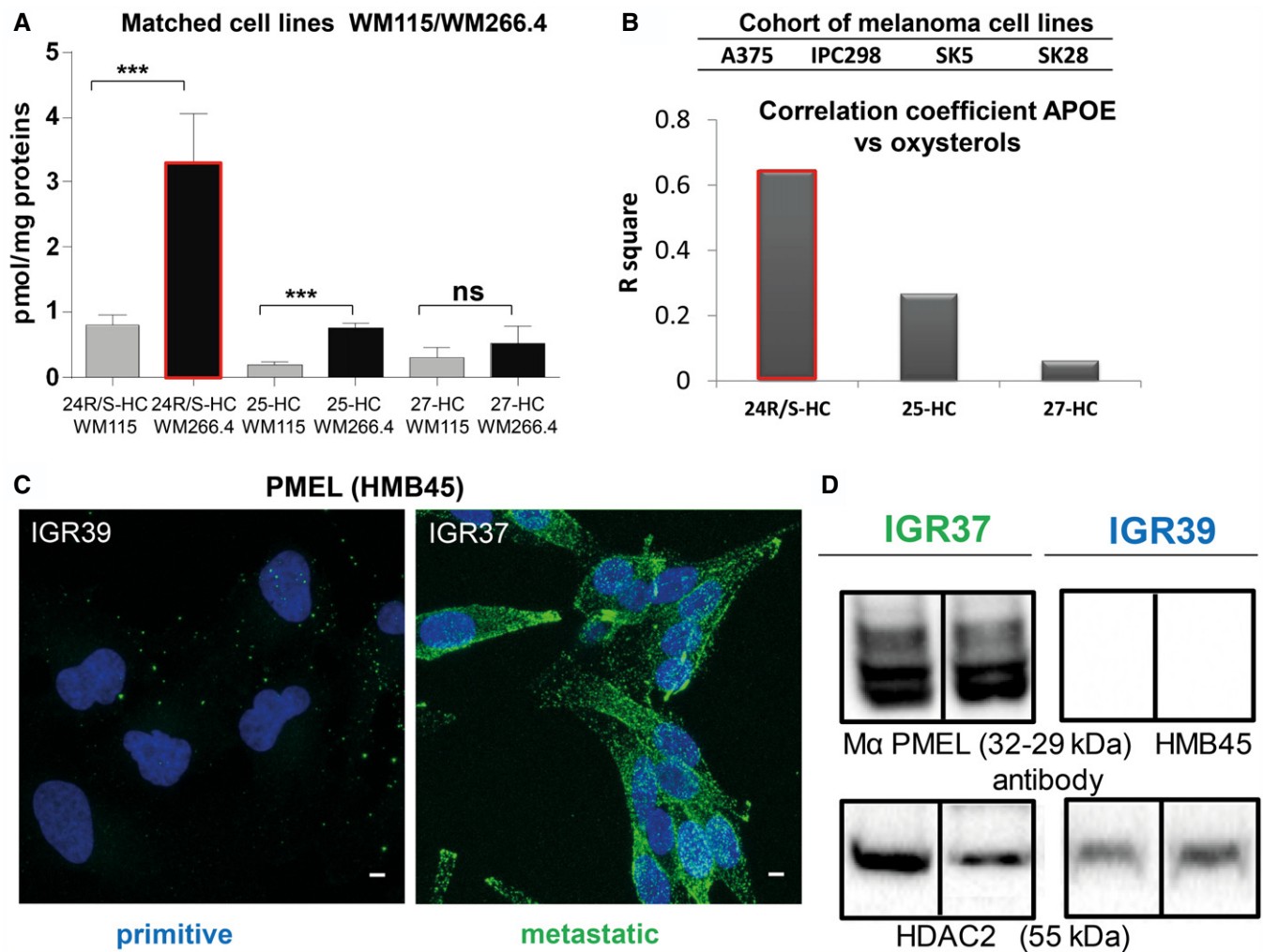

**Figure 3. Oxysterol quantification in primitive and metastatic melanoma cells and PMEL expression in IGR37 and IGR39 melanoma cell line.**

A  Absolute quantitation of 24-, 25- and 27-hydroxycholesterol in primitive and metastatic melanoma cells as indicated. ($T$-test analysis, ***$P < 0.001$, $N = 3$ biological replicates, data are mean ± SD).

B  Histogram representing R square of correlation analysis between absolute quantitation of 24-, 25- and 27-hydroxycholesterol and label free quantitation of APOE in a cohort of primitive (A375, IPC298) and metastatic (SKMEL5, SKMEL28) melanoma cell lines.

C  Confocal fluorescence images of anti-HMB45 PMEL antibody signal (green) and DAPI staining (blue) in IGR37 and IGR39. Scale bar is 10 μm.

D  Western blot on IGR37 and 39 cellular lysates probed with anti-HMB45 PMEL antibody and anti HDAC2 to check the loading of similar amount of total lysates.

Source data are available online for this figure.

higher mRNA levels of APOE and PMEL with respect to healthy donors (Fig EV3G). We therefore choose to pharmacologically inhibit BACE to test if it is actually involved in the formation of the protein aggregates that we observed in melanoma metastasis. NB-360 is a dual BACE inhibitor, known to impair the maturation into fibrils of both APP, in the central nervous system, and PMEL, in normal melanocytes (Neumann *et al*, 2015). We decided to use a dual BACE inhibitor as it has been observed that BACE1 is able to compensate for the function of BACE2 and vice-versa (Shimshek *et al*, 2016). Matched melanoma cell lines, i.e. IGRs, were treated with NB-360 at a concentration which is not cytotoxic. As shown in Fig EV3H (solid horizontal lines), cell viability is not affected upon NB360 treatment while the number of alive cells is decreased in both IGRs (Fig EV3H, histogram). Moreover, NB360 is able to

decrease the amount of melanin (Fig EV3I), indicating an impairment of PMEL amyloidogenic fragment formation (Shimshek *et al*, 2016). Successively, the secretome was analyzed by Secret3D (Dataset EV9, Peptide Atlas repository). Notably, primitive and metastatic secretome clustered separately, displaying a different profile which is coherent with the observation of a different phenotype; moreover, NB360 treatment affects both primitive and metastatic cells by targeting different proteins (Fig 5A, Dataset EV6). In particular, the amount of secreted PMEL decreased upon treatment in metastatic cells together with other amyloidogenic proteins and known BACE targets (Fig 5B). The overall impact of the drug was analyzed by performing pathway enrichment analysis. Upon treatment, the downregulated pathways were found to be linked to endocytosis, cell adhesion regulation, and ECM (Fig 5C). Among these

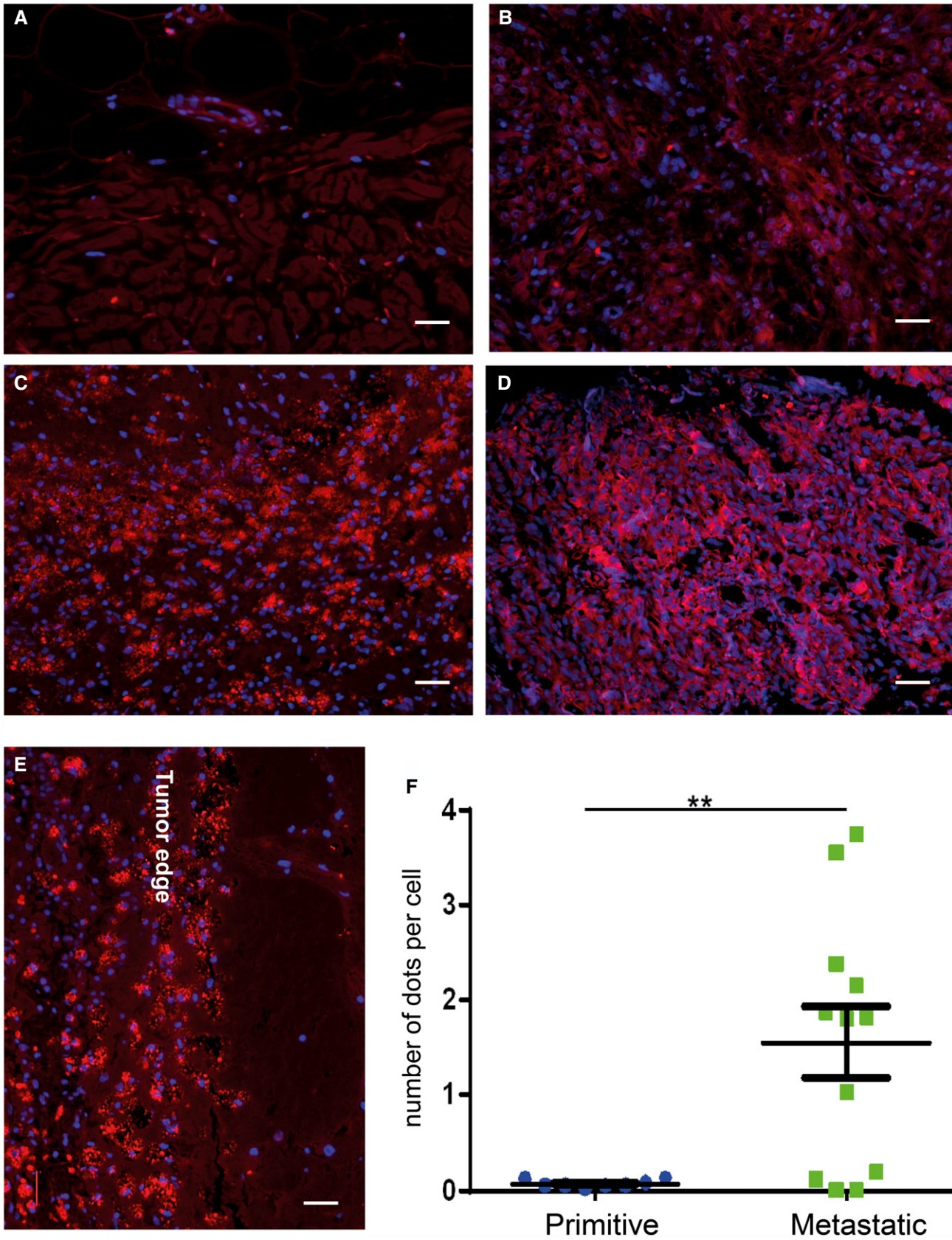

**Figure 4.**

**Figure 4. Protein aggregates accumulate in human metastatic melanoma.**

A–D   Immunofluorescence images with Proteostat (red) and DAPI (blue) staining on (A) human normal skin, (B) samples of primitive melanomas, (C) melanoma metastases in brain and (D) melanoma metastases in lung. Scale bar is 30 μm.

E   Details of brain metastases. Scale bar is 30 μm.

F   Quantitation of Proteostat-positive dots in primitive vs metastatic melanoma tissues: 6 tissues from metastatic lesions and 6 from primitive melanoma lesions were analyzed. For each tissue, two sections were quantified. *T*-test analysis was applied. *T*-test analysis, **$P < 0.01$ ($N = 6$, data are mean ± SEM).

pathways, we found that the majority of proteins affected by the treatment belong to the metastatic signature identified in the secretome (Fig 5B). Indeed, even if in both primitive and metastatic cells, the same pathways appear to be perturbed, we observed a stronger impact on the metastatic phenotype (Fig 5C). In particular, confocal microscopy analysis of Proteostat labeled cells after NB360 treatment showed a significant decrease of protein aggregates demonstrating that BACE is involved in their maturation into fibrils (Fig 5D and E).

### BACE2 inhibition impairs mechanotransduction in metastatic melanoma

Notably, among BACE-downregulated proteins, we identified Agrin (Dataset EV10). Agrin is a key protein that senses the extracellular stiffness and activates signaling events to induce the translocation of Yes-associated protein (YAP) into the nucleus (Chakraborty *et al*, 2017). YAP is a transcription factor that plays an important role in mechanotransduction along with the transcriptional co-activator with PDZ-binding motif (TAZ) (Dupont *et al*, 2011; Lamar *et al*, 2012). We postulated that protein aggregates in the extracellular space might activate mechanosignaling leading to YAP-mediated transcription. Endorsing our hypothesis, YAP nuclear localization was decreased in response to NB-360 treatment (Figs 5F and EV4A) and YAP target genes, e.g. CTGF, TFGBR2, IGBP4 and FZD1, were found to be downregulated by the drug in the metastatic secretome (Dataset EV6). YAP target gene, i.e. CTGF, is downregulated also at mRNA level upon NB360 treatment, attesting that YAP transcriptional activity is actually inhibited by the drug (Fig 5G).

Knowing that BACE1 and BACE2 are both involved in amyloid processing, we hypothesized that BACE2 should be mainly responsible for amyloid maturation in metastatic melanoma as it is more expressed in metastatic compared to primitive cells, while BACE1 shows an opposite behavior (Fig EV4B). Thus, we silenced BACE2 in metastatic IGR37 melanoma cells (iBACE2 KD IGR37), by using a pLKO-TET-on BACE2 shRNA (Fig EV4C). After doxycycline administration, BACE2 expression was reduced together with PMEL maturation into fibrils (Fig EV4D). Consequently, PMEL secretion was impaired together with other known BACE2 targets, i.e. SORT1 (Fig EV4E, Dataset EV11). Indeed, we found that more than 250 significantly regulated proteins were in common between BACE2 KD and NB360 treatment (Venn diagram in Fig EV4E) attesting that the pharmacological and the genetic approach shares a similar behavior. Moreover, we detected less protein aggregates by Proteostat staining (Fig EV4F) and we also confirmed that the YAP target CTGF was downregulated (Fig EV4G). Therefore, we can conclude that BACE2 KD effectively diminishes PMEL amyloid-like structures in the extracellular space and leads to YAP inactivation in metastatic melanoma cells. These findings were further confirmed by the specific BACE2 inhibitor 3I (BACE2 Ki = 1.6 nM; BACE1 Ki = 815.1 nM; Ghosh *et al*, 2019). Actually, 3I treatment of metastatic cells abrogates protein aggregates formation (Fig EV4H) and affects YAP transcriptional activity as measured by CTGF expression (Fig EV4I).

### PMEL amyloid fibrils induce mechanotransduction triggering the Agrin-pFAK-YAP axis

To get insights into the mechanism that links BACE2-dependent protein aggregates and YAP, we supplemented primary melanoma IGR39 cells with metastatic IGR37 conditioned medium and we measured the CTGF expression as exemplary of YAP target genes. As reported in Fig 5H, we detected an increased level of CTGF proving that the metastatic secretome is indeed able to modulate YAP activity. Furthermore, to investigate if the extracellular

**Figure 5. Secretome analysis of IGRs upon BACE inhibition.**

A   Unsupervised hierarchical clustering of the proteins identified and quantified in IGR37 and IGR39 upon treatment with DMSO or NB-360.

B   Volcano plot of the proteins secreted by IGR37 cells treated with DMSO or NB-360.

C   KEGG enrichment pathway analysis of the significantly regulated proteins upon BACE inhibition in both IGRs.

D   Confocal fluorescence images of Proteostat signal (1:2,000, red spots) and DAPI staining (blue), scale bar is 10 μm.

E   Quantitation of protein aggregates in IGRs by immunofluorescence analysis using Fiji software. (*T*-test analysis, **$P < 0.01$, $N = 3$ biological replicates, data are mean ± SD).

F   Quantitation, by immunofluorescence analysis, of YAP in different cellular compartments (*T*-test analysis, *$P < 0.05$, $N = 3$ biological replicates, data are mean ± SD). Images were quantified by subdividing cells into mostly cytosolic YAP (Cytosol), mostly nuclear YAP (Nuclear), or equal distribution (Total) from three biological replicates.

G   mRNA levels of CTGF measured by real-time PCR in IGR37 treated with DMSO or NB-360 (*T*-test analysis, *$P < 0.05$, $N = 3$ biological replicates, data are mean ± SD).

H   CTGF mRNA level measured by real-time PCR in IGR39 treated with IGR39 conditioned medium (CTRL) or with IGR37 conditioned medium (CM), $N = 4$ biological replicates. *T*-test analysis, ***$P < 0.001$, data are mean ± SD.

I   CTGF mRNA level measured by real-time PCR in IGR39 supplemented with recombinant PMEL amyloid fibrils (0.5 μM), $N = 3$ biological replicates. *T*-test analysis: ***$P < 0.001$, data are mean ± SD.

J   CTGF mRNA level measured by real-time PCR in IGR37 treated with DMSO, NB-360 or NB-360 plus recombinant PMEL amyloid fibrils (0.5 μM), $N = 3$ biological replicates. *T*-test analysis *$P < 0.05$, **$P < 0.01$. Data are mean ± SD.

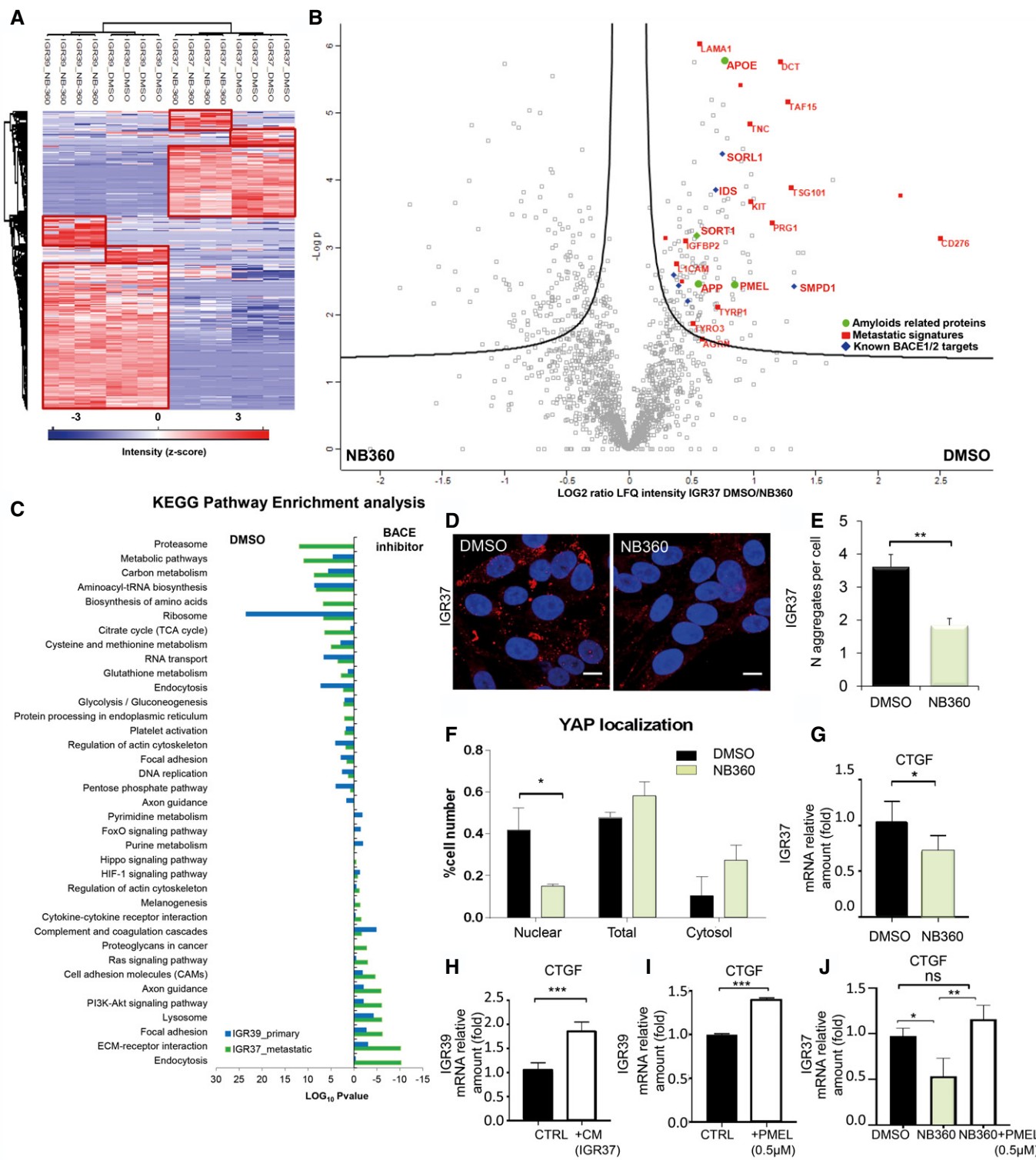

**Figure 5.**

amyloidogenic proteins act as "mechanotransducers" and are sufficient to activate YAP signaling, we exogenously added recombinant PMEL amyloid fibrils to the primary melanoma IGR39 cells. We choose PMEL because it is the most abundant amyloidogenic protein found in the secretome of metastatic melanoma cells (Fig 2E). Interestingly, PMEL fibrils alone increase CTGF expression

(Fig 5I) thus demonstrating that amyloids impinge on a signaling pathway able to activate YAP.

These data were further strengthened by a rescue experiment demonstrating that the administration of PMEL fibrils to NB360 treated cells is able to restore the expression of CTGF (Fig 5J). This experiment also indicates that, despite the broad effect of the dual

BACE inhibitor, PMEL fibrils alone impact on YAP transcriptional activity similarly to what happens when YAP is activated by canonical mechanotransduction signals. Furthermore, we also observed that PMEL fibrils promote Agrin expression and increase focal adhesion kinase (FAK) phosphorylation (Fig EV4J and K), indicating that extracellular amyloids are linked to mechanotransduction via FAK phosphorylation thus promoting the nuclear translocation and activation of YAP (Chakraborty *et al*, 2017).

### BACE inhibition impacts on proliferation and enhances chemo-sensitivity in melanoma cells

Convincing evidences indicate that mechanotransduction, through YAP activation, is able to affect cell proliferation and confer drug resistance to different chemical compounds (Oku *et al*, 2015).Therefore, we wondered if NB-360 might impact on metastatic proliferation and chemo-sensitivity. We evaluated the clonogenic activity of IGR melanoma cells upon treatment with NB360, and we found a diminished formation of new colonies and a decreased proliferation rate (Figs 6A and EV5A and B). This effect was similar in both primitive and metastatic cell lines. To address a preferential role of BACE2 in the metastatic phenotype, we used the selective BACE2 inhibitor 3I. Notably, 3I treatment impairs proliferation of metastatic cells while it has no effect on primary cells (Fig 6B). A similar behavior was also observed when BACE2 was genetically knocked down excluding off-target effects of the drug and demonstrating that the impairment of BACE2 expression directly impacts on cell proliferation in metastatic melanoma cells (Fig EV5C).

We then decided to test if BACE inhibition is also able to enhance the effect of conventional chemotherapy agents such as doxorubicin. Interestingly, the combination of the two drugs makes metastatic cells more sensitive to treatment (Fig 6C and D). Indeed, by evaluating the $IC_{50}$, we observed that the combined therapy of doxorubicin and NB360 is more efficient compared to doxorubicin alone (Fig 6E). We also tested if the combined therapy performed equally in WMs melanoma cell lines and we confirmed that, also in this case, the response to the combinatorial treatment had a more pronounced effect than treatment with doxorubicin alone (Fig EV5D).

## Discussion

Cross-talk between tumor cells and the microenvironment has recently gained increasing attention as it actively contributes to cancer progression and metastasis (Wang *et al*, 2017b). By performing system-level analysis on cellular models of primitive and metastatic phenotypes, we found the presence of protein aggregates in metastatic cells, both at cellular and extracellular levels. Secretome

analysis revealed that proteins involved in amyloid deposition are enriched in the metastatic microenvironment together with proteins involved in ECM scaffolding. Altered ECM is frequently observed in various cancers (Lampi & Reinhart-King, 2018) including melanoma (Miskolczi *et al*, 2018), where stiffening precedes disease development driving its progression through specific mechanical signaling (Pickup *et al*, 2014). We hypothesize that amyloidogenic proteins in the extracellular space might aggregate and that the deposition of such highly rigid material (Fitzpatrick *et al*, 2013) might activate signaling pathways in melanoma microenvironment. In accordance with our hypothesis, we found APOE as the most secreted protein in the metastatic cell lines. APOE is a lipoprotein, whose primitive function is transporting cholesterol, but it is also involved in the stabilization of amyloid-β fibrils in AD and of PMEL fibrils in melanocyte maturation (Bissig *et al*, 2016). Recently, APOE variants were found to be involved in melanoma progression and survival (Ostendorf *et al*, 2020). APOE expression is regulated by the nuclear LXR activation. In agreement, we showed higher level of the endogenous LXR agonist, i.e. 24-hydroxycholesterol in metastatic *versus* primitive cells. LXR activation was observed also in AD (Abildayeva *et al*, 2006), and recently, the involvement of oxysterols in tumor progression was reported also in melanoma (Ortiz *et al*, 2019). In our study, APOE was found associated with the secretion of proteins such as SORT1 (Carlo *et al*, 2013) and QPCT both involved in amyloid-β fibril stabilization (Morawski *et al*, 2014). Moreover, by using a specific dye, we observed the presence of protein aggregates both at cellular and extracellular levels. All these observations sustain our hypothesis that in metastatic melanoma extracellular environment there is an overproduction of amyloid-like structures. Despite melanoma progression is accompanied by cellular pigmentation (Kirkpatrick *et al*, 2006; Sarna *et al*, 2014), we found that also metastatic unpigmented cells, i.e. WM cell lines, actively secrete protein aggregates.

By analyzing tissues from melanoma patients, we highlighted the presence of protein aggregates also *in vivo*. According to our proteomic data, we observed amyloid-like protein aggregation enriched in metastatic lesions compared to primitive tumor tissues. Protein aggregates are hallmark of neurodegenerative disease such as AD, but their involvement in cancer progression is still poorly understood (Xu *et al*, 2011).

With the aim of understanding the biological relevance of these protein aggregates, we interfered with their production by targeting BACE2, the enzyme that assists the maturation and release of amyloidogenic peptides (Rochin *et al*, 2013). Interestingly, by interrogating TCGA and GTEX data, we found that BACE2 is highly expressed in melanoma patients compared to healthy donors, and its level of expression correlates with poor prognosis. Moreover, in IGRs cell lines we found that BACE2 was more expressed in metastatic compared to primitive cells. By using a BACE inhibitor or by

---

**Figure 6. BACE inhibition affects proliferation and chemo-sensitivity in melanoma cells.**

A  MTT assay for IGRs treated with DMSO or NB-360 (25 μM). *N* = 4 biological replicates. *T*-test analysis **$P < 0.01$, ***$P < 0.001$. Data are mean ± SD.

B  MTT assay for IGRs cells treated with DMSO or inhibitor 3I (3 μM). *N* = 3 biological replicates. *T*-test analysis **$P < 0.01$. Data are mean ± SD.

C, D  MTT assay of IGRs treated with NB-360 (25 μM) and different concentration of doxorubicin as indicated. *N* = 4 biological replicates. *T*-test analysis: $0.01 < *P < 0.05$, **$P < 0.01$. Data are mean ± SD.

E  *T*-test analysis of IC50 values for doxorubicin used alone or in combination with NB-360 (combo). Data are mean ± SD. *T*-test analysis: $0.01 < *P < 0.05$; $0.001 < **P < 0.01$.

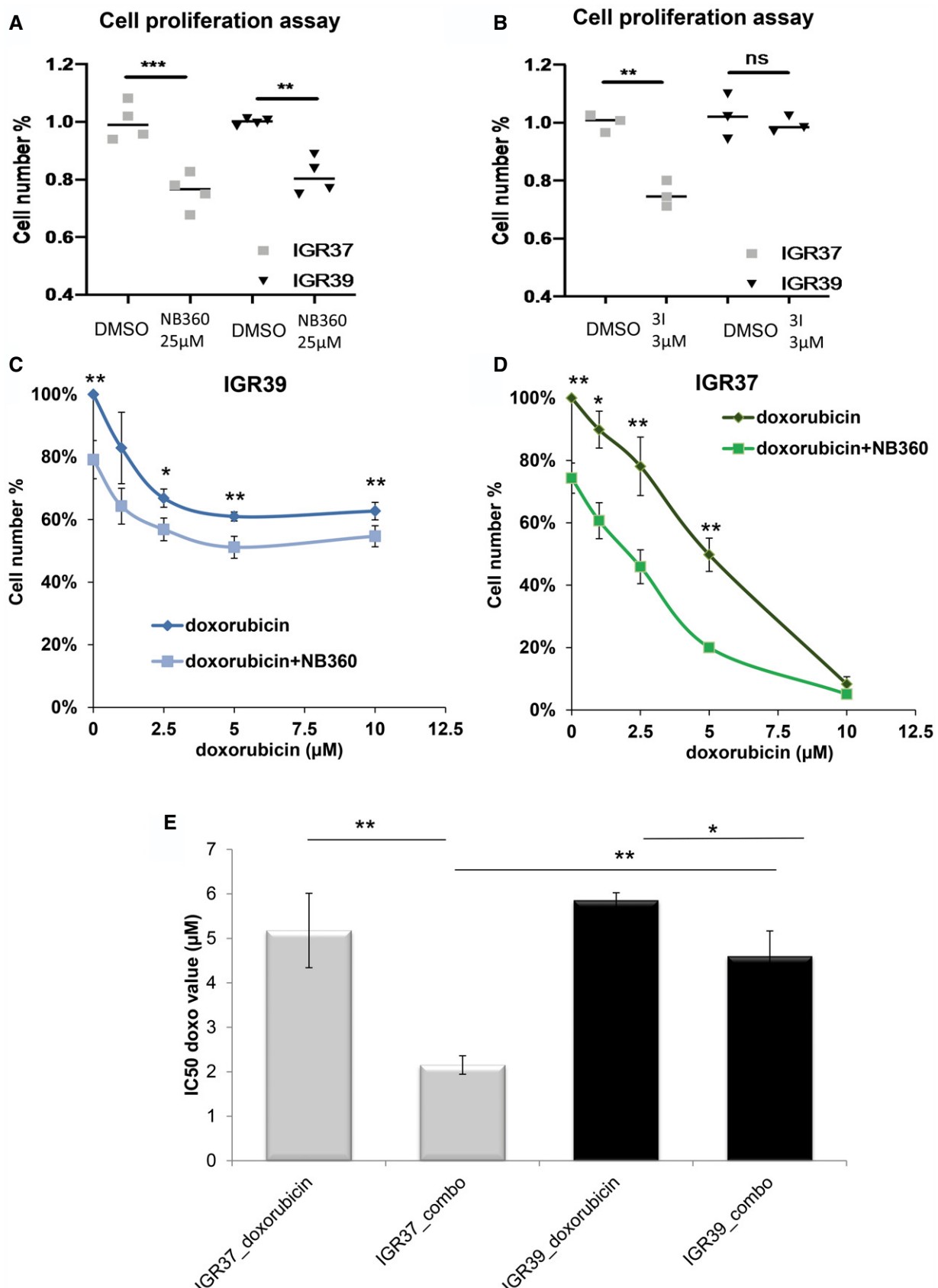

Figure 6.

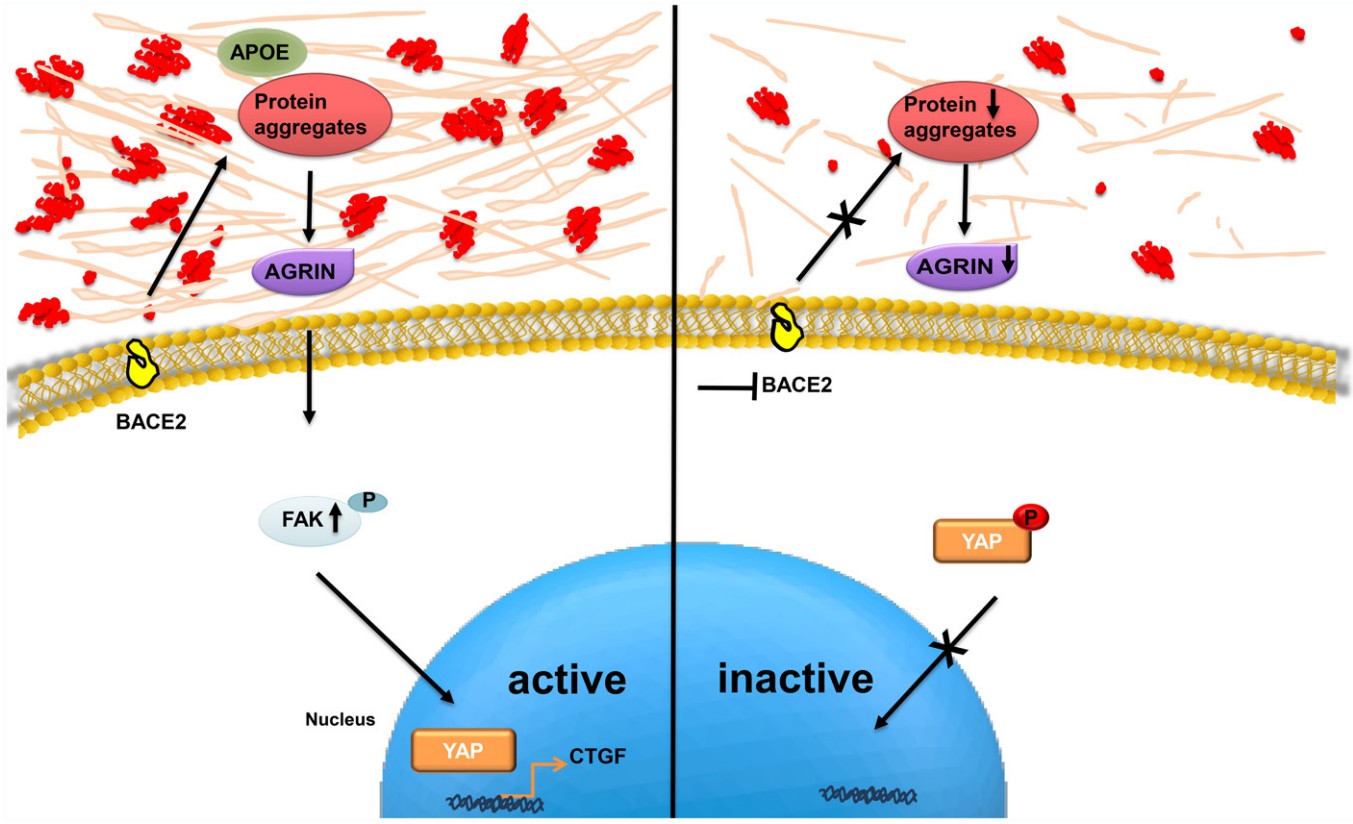

**Figure 7. BACE as a new regulator of YAP in metastatic melanoma cells.**

In metastatic melanoma, BACE2 activity assists the secretion of protein aggregates into the extracellular space. The presence of these aggregates might be sensed by Agrin, known to activate the YAP signaling cascade, and is able to induce YAP-mediated CTGF transcription. In turn, melanoma cells treated with BACE inhibitors produce fewer protein aggregates and show a lower YAP transcriptional activity.

genetically impairing BACE2 expression, we reduced the formation of protein aggregates, impaired PMEL shedding, but also affected the secretion of APOE, SORT1, and proteins of the extracellular matrix, as Agrin. In AD, Agrin co-localizes with amyloid plaques and stabilizes amyloid-β fibrils (Cotman *et al*, 2000). Agrin is also a mechanical sensor that transduces ECM rigidity signals by inducing Yes-associated protein (YAP) activation (Chakraborty *et al*, 2017). Over the past decade, YAP has emerged as important driver of cancer development (Lamar *et al*, 2012). It has been postulated that YAP contributes to both invasive and metastatic behavior in melanoma (Nallet-Staub *et al*, 2014), encouraging researchers to target its activity for anti-cancer therapy (Johnson & Halder, 2014). In the present work, we demonstrated that amyloid fibrils in metastatic melanoma induce YAP nuclear localization and its transcriptional activity. Moreover, we also demonstrated that PMEL fibrils are sufficient to promote YAP activation and to increase Agrin expression and FAK phosphorylation, which are known and well-recognized markers of mechanotransduction (Chakraborty *et al*, 2017). Mechanistically, we propose a model where BACE2 assists the maturation of protein aggregates, which accumulate in the extracellular space where they could be sensed by Agrin leading to YAP activation (Fig 7). We cannot exclude that amyloid fibrils have also other effects but we clearly show that they impinge on the mechanotransduction pathway. Further work is necessary to clarify if this is a direct link or if and which other molecules are required.

In melanoma, YAP overexpression confers resistance to BRAF inhibitor, whereas YAP depletion increases drug sensitivity (Kim *et al*, 2016). Consistently, we observed that melanoma cells treated with BACE inhibitor become less proliferative and more sensitive to chemotherapy. Our data are consistent with a recent study where BACE inhibition was able to impair melanoma brain metastatic proliferation *in vivo* (preprint: Kleffman *et al*, 2019). Indeed, we found that a combined treatment of doxorubicin and NB360 is more effective than doxorubicin alone. Actually, protein aggregates rigidity might contribute in the formation of the "safe haven", favoring tumor growth and melanoma resistance (Hirata *et al*, 2015). Supporting this hypothesis, rigid tumor microenvironment was often associated with the formation of a physical barrier affecting drug uptake (Holle *et al*, 2016). Moreover, coherently with our model, Wang *et al* reported that PMEL silencing affects mouse melanoma cells' proliferation similarly to what we observed by BACE inhibition (Wang *et al*, 2017a), while PMEL expression is associated with resistance to chemotherapy in melanoma (Johansson *et al*, 2013). In particular, siRNA inhibition of PMEL in the MNT-1 melanoma cells sensitized the cells to both paclitaxel and cisplatin in line with the increased chemo-sensitivity induced by BACE inhibition described in our work.

Melanoma recent therapies are remarkably efficient in a subpopulation of patients; for those who do not respond though, melanoma remains a devastating disease raising the need of alternative

therapies. Here, we found a potential new druggable target, i.e. BACE, able to affect melanoma microenvironment. Targeting BACE in combination with chemotherapy might open new ways to counteract metastatic melanoma. Moreover, this therapy might also interfere with the mechanosignaling pathway that can promote metastatic growth and survival (Lamar *et al*, 2012). In our work, we have underlined a cell autonomous effect of protein aggregates deposition, but it would be interesting to explore the effect of the presence of amyloid-like structures also on neighboring cells. It has been recently demonstrated that *in vivo* ECM production is mostly fibroblastic dependent, while ECM remodeling is both tumor cell and fibroblastic cell dependent (Lamar *et al*, 2012). Here, we provided evidences that amyloid aggregates are secreted by melanoma cells and might contribute to ECM remodeling. Moreover, as amyloidogenic protein overexpression has been reported also in other tumor types, such as breast (Danish Rizvi *et al*, 2018) and pancreas (Westermark *et al*, 2017), it is attractive to think that the same mechanism could be exploited also in other diseases.

# Materials and Methods

## Reagents and Tools table

| Reagent or Resource | Source | Identifier |
|---|---|---|
| **Antibodies** | | |
| Mouse monoclonal HMB-45 | Thermo Fisher | Catalog # MA5-13232 |
| Mouse monoclonal anti-YAP | Santa Cruz | sc-101199 |
| Rabbit polyclonal anti-HDAC2 | Abcam | Ab16032 |
| Mouse monoclonal anti-FAK | Santa Cruz | sc-271195 |
| Mouse monoclonal anti-pFAK (Y397) | Santa Cruz | sc-81493 |
| Mouse monoclonal anti-Actin | Santa Cruz | sc-47778 |
| **Chemicals, Peptides, and Recombinant Proteins** | | |
| 25-hydroxycholesterol (D6) | Avanti Polar Lipids | LM-4110 |
| 24(R/S)-hydroxycholesterol (D6) | Avanti Polar Lipids | LM-4110 |
| 27-hydroxycholesterol (D6) | Avanti Polar Lipids | LM-4114 |
| NB-360 | Novartis | N/A |
| Doxorubicin | Selleck Chemicals | #S1208 |
| ECL | Amersham | RPN2232 |
| DAPI | Sigma-Aldrich | CAS Number 28718-90-3 |
| MUSE | Millipore | Catalogue Number MCH100102 |
| Proteostat | Enzo Life science | ENZ-51023 |
| DMSO | Euroclone | EMR385258 |
| **Critical Commercial Assays+40:64** | | |
| Bradford | Applied Biosystems | Cat#4368814 |
| LightCycler® 480 SYBR Green I Master | Roche | Product No. 04707516001 |
| SuperScript™ VILO™ cDNA Synthesis Kit | Invitrogen | Cat. No. 11754-050 and 11754-250 |
| Ni-NTA Agarose | Qiagen | Cat No./ID: 30230 |
| **Experimental Models: Melanoma Cell Lines** | | |
| IGR37 | DSMZ | ID ACC 237 |
| IGR39 | DSMZ | ID ACC 239 |
| WM115 | IZSBS | ID BS-TCL74 |
| WM266.4 | ATCC | ID CRL-1676 |
| A-375 | IZSBS | ID BS-TCL88 |
| C32 | IZSBS | ID BS-TCL150 |
| IPC-298 | DSMZ | ID ACC 251 |
| SK-MEL-5 | NCI-60 | ID 0507403 |
| MeWo | ICLC | ID HTL97019 |

**Reagent and Tools table** (continued)

| Reagent or Resource | Source | Identifier |
| --- | --- | --- |
| Sk-MEL-28 | NCI-60 | ID 0507398 |
| HEK293T | ICLC | ID HTL04001 |
| **Software and Algorithms** | | |
| GraphPad Prism software | GraphPad Software | http://www.graphpad.com |
| Microsoft Excel 2010 | Microsoft Office | N/A |
| Fiji | ImageJ Software | https://imagej.net/Fiji |
| MaxQuant | N/A | http://www.coxdocs.org/doku.php?id=maxquant: common:download_and_installation |
| Perseus | N/A | http://www.coxdocs.org/doku.php?id=perseus:start |
| Venny | N/A | http://bioinfogp.cnb.csic.es/tools/venny/ |
| **Deposited data** | | |
| .RAW files of the proteomic data were deposited together with all peptides identified and parameters used for the analysis | Peptide Atlas repository | PASS01358 |

## Methods and Protocols

### Cell culture

Human melanoma cell lines such as IGR37, IGR39, and IPC-298 were purchased from DSMZ; MEWO cell line was purchased from ICLC; SK-MEL-5 and SK-MEL-28 were purchased from NCI-60; WM266.4 was purchased from ATCC; WM115, A-375 and C32 were purchased from IZSBS; and HEK293T cells were purchased from ICLC.

All the cells were cultured in Dulbecco modified Eagle's medium (DMEM) + 10% FBS S.A. + 2 mM L-Glutamine except for IPC-298 that was cultured in RPMI-1640 + 10% FBS S.A. + 2 mM L-Glutamine. Cell lines were tested for mycoplasma by mycoplasma PCR Test Kit.

### Analysis of human biopsies

Formalin-fixed paraffin-embedded tissues were sliced into serial 8-μm-thick sections and collected for immunohistochemical (IHC) staining. Human paraffin samples were stained for Haematoxylin/Eosin (Diapath) to assess histological features, according to standard protocol. For Ki67 immunoanalysis, paraffin was removed with xylene and sections were rehydrated in graded alcohol. Tissue slides were incubated in 10% peroxidase solution for 1 h at 65°C to remove melanin pigments, and then, antigen retrieval was carried out using preheated target retrieval solution for 45 min at 95°C. Tissue sections were blocked with FBS serum in PBS for 60 min and incubated overnight with primary antibody (Thermo Scientific, 1:50). The antibody binding was detected using a polymer detection kit (GAR-HRP, Microtech) followed by a diaminobenzidine chromogen reaction (Peroxidase substrate kit, DAB, SK-4100; Vector Lab). All sections were counterstained with Mayer's hematoxylin, mounted in Eukitt (Bio-Optica), and then visualized with an Olympus BX51 or an Olympus BX63 upright widefield microscope using NIS-Elements (Nikon, Tokyo, Japan) or MetaMorph 7.8 software (Molecular Devices, San Jose, CA, USA), respectively. For Proteostat aggresome detection, deparaffinized and rehydrated slides were fixed in 4% PFA for 15 min, incubated in Proteostat solution (1:2,000, Proteostat Aggresome Detection Kit, Enzo) for 3 min, and then destained in 1% acetic acid for 20 min at room temperature. To visualize the cell nuclei, human slides were counterstained with 4,6-diamidino-2-phenylindole (DAPI, Sigma-Aldrich), mounted with a phosphate-buffered saline/glycerol solution, and examined with confocal or widefield microscopy. Confocal microscopy was performed on a Leica TCS SP5 confocal laser scanning based on a Leica DMI 6000B inverted motorized microscope. The images were acquired with a HC FLUOTAR L 25X/NA0.95 VISIR water immersion objective using the 405 nm and the 488 nm laser lines. The software used for all acquisitions was Leica LAS AF. Widefield microscopy was performed on an Olympus BX63 upright microscope equipped with a motorized stage for mosaic acquisitions and with both Hamamatsu ORCA-AG black and white camera and Leica DFC450C color camera. The mosaic images were acquired using a UPlanSApo 4X/NA0.16 dry objective with MetaMorph 7.8 software (Molecular Devices). Quantitative analysis of stained signals was performed using ImageJ software. Both protein aggregates (dots) and nuclei were analyzed. Dots were normalized on nuclei, and $T$-test statistical analysis was performed to estimate the differences between primary ($N = 6$) and metastatic tissues ($N = 6$). All the analysis was done in technical duplicates.

### Time-lapse microscopy

For live cell imaging experiments, melanoma cells were cultured in six-well plates ($5 \times 10^3$ cells per plate). Cultures were transferred to a live cell imaging workstation composed by an Olympus IX81 inverted microscope equipped with motorized stage and a Hamamatsu ORCA-Flash4.0 camera. The images were collected every 5 min for a total recording time of 72 h for each dish using a LUC Plan FLN 20X/NA0.45 Ph dry objective with CellSens software (Olympus). The analysis was done, in biological triplicate, by using trackmate (Fiji).

### Secretome preparation from cell cultures and SILAC labeling

All melanoma cells were grown in a DMEM except for IPC-298 which was grown in a RPMI medium, complemented with essential amino acids Arg and Lys, containing naturally occurring atoms (Sigma) (the light medium) or two of their stable-isotope counterparts (the medium and heavy media) (Cambridge Isotope

Laboratories, Inc.; CIL). The medium culture contained arginine (L-Arg $^{13}C_6$-$^{14}N_4$) and lysine (L-Lys $^{13}C_6$-$^{15}N_2$), and the heavy culture contained arginine (L-Arg $^{13}C_6$-$^{15}N_4$) and lysine (L-Lys $^{13}C_6$-$^{15}N_2$) amino acids. After five cell divisions to obtain full incorporation of the labeled amino acids into the proteome, cells were counted and equal numbers of cells were split to 15-cm dishes at roughly 50% confluence. Once cell lines reached ~70% confluence, one 15-cm dish of each cell line was washed 3 × with PBS and 3 × with serum-free media. Cells were starved in serum-free media for 18 h, and the conditioned media (CM) were centrifuged (800 $g$, 3 min), filtered (0.22 μm) to remove detached cells, and concentrated via centrifugation at 4,500 $g$ in 10 kDa molecular weight cutoff concentrating columns. Then, 500 μl of concentrated medium was filtered by microcon filters with 10 kDa cutoff (Millipore) and buffer was exchanged with 8 M Urea 100 mM Tris or PBS.

### Protein aggregates detection

Aggregates Fluorescence measurement was performed following manufacturer's instructions (http://www.enzolifesciences.com/fileadmin/files/manual/ENZ-51023_insert.pdf). Briefly, 2 μl of the diluted PROTEOSTAT®Detection Reagent was added into the bottom of each well of a 96-well microplate. 98 μl of the secreted proteins in PBS was added to each well. Protein concentration was 10 μg/ml. The final concentration of the PROTEOSTAT detection dye was 1,000-fold dilution in the assay. We run control samples as well as 1× Assay Buffer alone (no protein), as a blank sample. The microplate containing samples was incubated in the dark for 15 min at room temperature. Generated signals were read with a fluorescence Microplate Reader (Tecan Infinite 200) using an excitation setting of about 550 nm and an emission filter of about 600 nm.

### Secretome analysis

Proteins secreted by $2 × 10^6$ cells were replaced in 8 M Urea 100 mM Tris pH 8 and sonicated with BIORUPTOR (3 cycles: 30 s on/30 s off). By using microcon filters with 10 kDa cutoff (Millipore), cysteine reduction and alkylation were performed adding 10 mM TCEP (Thermo scientific) and 40 mM 2-Chloroacetamide (Sigma-Aldrich) in 8 M Urea 100 mM Tris pH 8 for 30 min at room temperature, as described for the FASP protocol. Buffer was exchanged by centrifugation at 9,300 $g$ for 10 min, and PNGase F (New England Biolabs) (1:100 = enzyme: secreted proteins) was added for 1 h at room temperature following manufacturer's instruction. Buffer was again exchanged by centrifugation at 9,300 $g$ for 10 min with 50 mM ammonium bicarbonate, and proteins were in solution digested with trypsin (Trypsin, Sequencing Grade, modified from ROCHE) (1:50 = enzyme: secreted proteins) overnight at 37°C. Peptides were recovered on the bottom of the microcon filters by centrifugation at 9,300 $g$ for 10 min and on the top, adding two consecutive wash of 50 μl of 0.5 M NaCl. The undigested polypeptides on the top of the filters were further digested with GluC (Endoproteinase Glu-C Sequencing Grade ROCHE) (1:50 = enzyme: secreted proteins) overnight at 37°C upon buffer exchange with phosphate buffer (pH 7.8). Eluted peptides were purified on a C18 StageTip. 1 μg of digested sample was injected onto a quadrupole Orbitrap Q-exactive HF mass spectrometer (Thermo Scientific). Peptide separation was achieved on a linear gradient from 95% solvent A (2% ACN, 0.1% formic acid) to 55% solvent B (80% acetonitrile, 0.1% formic acid) over 75 min and from 55% to 100%

solvent B in 3 min at a constant flow rate of 0.25 μl/min on UHPLC Easy-nLC 1000 (Thermo Scientific) where the LC system was connected to a 23-cm fused-silica emitter of 75 μm inner diameter (New Objective, Inc. Woburn, MA, USA), packed in-house with ReproSil-Pur C18-AQ 1.9 μm beads (Dr Maisch Gmbh, Ammerbuch, Germany) using a high-pressure bomb loader (Proxeon, Odense, Denmark).

The mass spectrometer was operated in DDA mode as described previously (Matafora *et al*, 2017): dynamic exclusion enabled (exclusion duration = 15 s), MS1 resolution = 70,000, MS1 automatic gain control target = 3 × 106, MS1 maximum fill time = 60 ms, MS2 resolution = 17,500, MS2 automatic gain control target = 1 × 105, MS2 maximum fill time = 60 ms, and MS2 normalized collision energy = 25. For each cycle, one full MS1 scan range = 300–1,650 m/z was followed by 12 MS2 scans using an isolation window of 2.0 m/z.

All the proteomic data as raw files, total proteins, and peptides identified with relative intensities and search parameters have been loaded into Peptide Atlas repository (PASS01358).

### MS analysis and database search

MS analysis was performed as reported previously (Matafora *et al*, 2015). Raw MS files were processed with MaxQuant software (1.5.2.8), making use of the Andromeda search engine (Cox *et al*, 2011). MS/MS peak lists were searched against the UniProtKB Human complete proteome database (uniprot_cp_human_2015_03) in which trypsin and GluC specificity was used with up to two missed cleavages allowed. Searches were performed selecting alkylation of cysteine by carbamidomethylation as fixed modification, and oxidation of methionine, N-terminal acetylation and N-Deamination as variable modifications. Mass tolerance was set to 5 ppm and 10 ppm for parent and fragment ions, respectively. A reverse decoy database was generated within Andromeda, and the false discovery rate (FDR) was set to < 0.01 for peptide spectrum matches (PSMs). For identification, at least two peptide identifications per protein were required, of which at least one peptide had to be unique to the protein group.

### Quantification and statistical analysis

SILAC and Label free from DDA.raw files were analyzed by MaxQuant software for protein quantitation, and depending from the experiment, SILAC Ratio or LFQ intensities were used. Statistical analysis was performed by using Perseus software (version 1.5.6.0) included in MaxQuant package. *T*-test and ANOVA statistical analysis were performed applying FDR < 0.05 or $P < 0.05$ as reported. KEGG enrichment pathway analysis was performed via EnrichR (http://amp.pharm.mssm.edu/Enrichr), using the Gene ID of the identified proteins.

### Oxysterol quantification

Oxysterols were prepared from melanoma cell lines using a modified version of the protocol described by Griffiths *et al* (2013), consisting in an alcoholic extraction and a double round of reverse-phase (RP) solid-phase extraction (SPE; Soncini *et al*, 2016). Briefly, melanoma cell pellets ($2 × 10^6$ cells) were sonicated for 5 min by adding 1.0 ml of ethanol supplemented with + 20 pmol of each deuterated standard. 400 μl $H_2O$ was added and sonicated for 5 min (final volume 1.5 ml of 70% ethanol). Upon centrifugation at

14,000 *g*, 4 °C for 30 min, the supernatant was collected. The extract was applied to a preconditioned Sep-Pak tC18 cartridge (Waters). The oxysterol-containing flow-through was collected, together with the first 70% (vol/vol) ethanol wash. The collected oxysterols were vacuum-evaporated and reconstituted in 100% (vol/vol) isopropanol, diluted in 50 mM phosphate buffer, and oxidized by cholesterol oxidase addition. The reaction was stopped by methanol. Reactive oxysterols were then derivatized by Girard P reagent (TCI Chemical) and further purified by reverse-phase chromatography using a Sep-Pak tC18 cartridge to eliminate the excess of GirardP reagent. Purified oxysterols were diluted in 60% (vol/vol) methanol and 0.1% formic acid. Eight µl of sample was resolved by on a nano-HPLC system connected to a 15-cm fused-silica emitter of 75 µm inner diameter (New Objective, Inc. Woburn, MA, USA), packed in-house with ReproSil-Pur C18-AQ 1.9 µm beads (Dr Maisch Gmbh, Ammerbuch, Germany) using a high-pressure bomb loader (Proxeon, Odense, Denmark). It was used a 12-min gradient from 20% to 100% of solvent B [63.3% (vol/vol) methanol, 31.7% (vol/vol) acetonitrile, and 0.1% formic acid], where solvent A is composed of 33.3% methanol, 16.7% acetonitrile, and 0.1% formic acid. Eluting oxysterols were acquired on a quadrupole Orbitrap Q-Exactive HF mass spectrometer (Thermo Scientific), where the survey spectrum was recorded at high resolution ($R = 140,000$ at 200 m/z) and the five most intense peaks were further fragmented. The identification of the oxysterols species was made by comparing the retention times of the analytes with those of the synthetic, deuterated standards previously run on the same system in the same chromatographic conditions. The quantification was achieved by means of stable-isotope dilution MS using internal standards. The total ion current for derivatized oxysterols was extracted for each acquisition, areas of the peaks were integrated manually using Xcalibur software, and the absolute amount of oxysterols was determined by comparing their areas with those of the internal standards, using the following equation:

$$\mathrm{Conc}_x = (\mathrm{I}_x/\mathrm{I}_{strd}) \times \mathrm{Conc}_{strd}$$

### Protein quantification
Protein quantification was performed using Bradford assay (Bio-Rad). For each sample, the absorbance was measured by a spectrophotometer at a wavelength of 595 nm. Sample protein concentration was determined based on a bovine serum albumin (BSA) standard curve.

### Western blot assays
For Western blot analyses, proteins were extracted in buffer containing 8 M Urea, 100 mM Tris–HCl pH 8. Briefly, cell lysates (50 µg) were separated by SDS–PAGE using a precast polyacrylamide gel with a 4% to 12% gradient (Invitrogen). After the electrophoretic run, proteins were transferred onto a 0.22 µm nitrocellulose membrane (Amersham Protran, GE Healthcare) in wet conditions. The assembled sandwich was loaded in a Trans-Blot Cell (Bio-Rad) and immersed in 1× cold Tris-Glycine transfer buffer with the addition of 20% methanol. The transfer was allowed overnight at constant voltage (30 V). Correct protein transfer was verified staining the membrane with Ponceau red (Sigma-Aldrich) for few seconds. After washing the membrane with Tris-buffered Saline-Tween 20 (TBST, 1× TBS with 0.1% Tween-20), non-specific

binding of antibodies was blocked by adding 5% low-fat dry milk in TBST for 1 h at room temperature. Murine Anti-human Pmel17 (HMB45 Thermo scientific) primary monoclonal antibody was diluted in the same blocking solution to a final concentration of 1:100, mouse anti-FAK(1:500) (Santa Cruz), mouse anti-pFAK Y397 (1:300) (Santa Cruz) The anti-HDAC2 antibody (Cell Signalling) and anti-Actin antibody (Santa Cruz) were used to normalize the amount of proteins loaded onto the gel. Anti-murine IgG1 secondary antibody conjugated with the enzyme horseradish peroxidase (HRP) was used to a final concentration of 1:2,000 in 5% milk-TBST.

### Immunofluorescence
Cells were fixed and permeabilized as describe previously (Matafora *et al*, 2009). After treatment, cells were fixed with 4% (wt/vol) paraformaldehyde, blocked with PBS-BSA (1% wt/vol), made permeable with Triton X-100 0.2% (Sigma-Aldrich) for 3 min, and incubated with Proteostat (1:1,000) or specific antibodies diluted in 0.2% bovine serum albumin in PBS. Cells were then washed three times with PBS and stained with DAPI (Sigma-Aldrich). Cells were observed by confocal microscopy performed on a Leica TCS SP5 or a Leica TCS SP2 AOBS confocal laser scanning. The confocal systems were, respectively, based on a Leica DMI 6000B or a DM IRE2 inverted microscope equipped with motorized stage. The images were acquired with an HCX PL APO 63X/NA1.4 oil immersion objective using the 405 nm, 488 nm, or 561 nm laser lines. The software used for all acquisitions was Leica LAS AF (on TCS SP5 system) or Leica Confocal Software (on TCS SP2 AOBS System).

### Recombinant PMEL (rMα) expression and purification
The luminal fragment of PMEL, rMα, consisting of amino acids 25–467 was subcloned from PGEX vector into a pET28a vector, in order to have 6xHis tag at the N-terminus, and expressed in BL21-DE3 *Escherichia coli*. Shaken cultures were grown at 37°C to $OD_{600} = 0.5$ in the presence of kanamycin and then induced with 1 mM IPTG for 4 h. Cells were collected via centrifugation at 4°C, resuspended in TBS (Tris-buffered saline: 150 mM NaCl, 50 mM Tris–HCl, pH 7.6), and frozen at −80°C. The resuspended pellet was thawed, and the cells were lysed by probe sonication. rMα formed inclusion bodies that were collected by centrifugation after three washings in 1.5 M NaCl, 100 mM Tris–HCl pH 7.4, 1% Triton X-100 buffer and two in TBS (Fowler *et al*, 2006). The inclusion bodies were dissolved in 9 M Urea, 100 mM $NaH_2PO_4$, 10 mM Tris–HCl pH 8.0 and then filtered through a 0.22 µm cellulose acetate filter and stored at room temperature. The protein was purified using Ni-NTA agarose beads (Qiagen, Germany) under denaturing condition. Briefly, 2 ml 50% slurry of Ni-NTA agarose beads were equilibrated with binding buffer (9 M Urea, 100 mM $NaH_2PO_4$, 10 mM Tris–HCl pH 8.0) before adding the sample. After binding for 1 h and 30 min, two washes were performed with 9 M Urea, 100 mM $NaH_2PO_4$, 10 mM Tris–HCl pH 6.5; elution was obtained in 9 M Urea, 100 mM $NaH_2PO_4$, 10 mM Tris–HCl pH 4.5.

### PMEL aggregates refolding and administration to cells
Recombinant PMEL aggregates refolding was obtained by slightly modifying a previously described protocol (Fowler *et al*, 2006). In particular, we did sequential dilutions form denaturing to native condition by performing first gel filtration and then buffer

exchange. Briefly, after Ni-NTA purification, recombinant PMEL was subjected to gel filtration in mild denaturing buffer (4 M Urea, 100 mM Tris–HCl pH 8.0) on a Superdex 200 16/60 column (GE Healthcare Life Sciences, USA), in order to allow partial refolding and avoid the elution of the protein in the void volume. The fractions corresponding to PMEL elution were pulled together and concentrated by using Amicon Ultra centrifugal tubes with 10 kDa cutoff (Millipore, USA). To allow a complete refolding and cell culture treatment, the buffer was exchanged with PBS. Recombinant PMEL aggregates were administered to cells in culture media at a final concentration of 0.5 μM.

### IGR39 treatment with IGR37 conditioned medium
IGR39 were seeded at the concentration of 100,000 cells/well. At the same time, IGR37 and IGR39 were seeded at 60% confluency in a 10-cm petri dish. After 24 h, the media deriving from IGR37 and from IGR39 were filtered on a 0.22 μm cellulose acetate filter, in order to remove dead cells and cells debris, and administered to IGR39. After 24 h, cells were harvested and RNA extraction was performed.

### RNA extraction, RT–PCR and real-time PCR
Total RNA was extracted using Maxwell RSC simply RNA (Promega, USA) according to manufacturer's instructions, and RNA was quantified by nanodrop. 1 μg of total RNA was used for retro-transcription using SuperScript™ VILO™cDNA Synthesis Kit (Invitrogen, USA). cDNA was diluted 1:10, and qPCR was performed using LightCycler® 480 SYBR Green I Master (Roche, Switzerland). The primer sequences are provided below. Expression data were normalized to the geometric mean of the housekeeping gene RPLP0 to control the variability in expression levels and were analyzed using the $2^{-\Delta\Delta CT}$ method. Primers for qPCR: CTGF-Forward primer: GGGAAATGCTGCGAGGAGT, CTGF-Reverse primer: GCCAAACTGT CTTCCAGTC; AGRN-Forward primer: TTGTCGAGTACCTCAA CGCT, AGRN-Reverse primer: CAGGCTCAGTTCAAAGTCGT; RPLP0-Forward primer: GTTGCTGGCCA ATAAGGTG, RPLP0-Reverse primer: GGGCTGGCACAGTGACTT.

### Cell viability assays
Melanoma cell lines were seeded into 6-well plates. MUSE reagent was added to detached cells, and cell viability was assessed according to the manufacturer's instructions (http://www.merckmilli pore.com/IT/it/product/Muse-Count-Viability-Assay-Kit-100-Tests, MM_NF-MCH100102#anchor_UG). Viability was accessed by measuring cell confluence (%) and number of dead and alive cells by using Muse™ Cell Analyzer.

### MTT cell viability assay
To perform 3-(4,5-dimethylthiazol-2-yl)-2,5-diphenyltetrazolium bromide (MTT; Sigma) cell viability assay, melanoma cells were seeded in 96-well plates ($5 \times 10^3$ cells/well) and were treated with 3I, doxorubicin or/and NB360 as indicated in the text. At the end of the experiments, the cell cultures were supplemented with 150 μl of 0.5 mg/ml MTT assay and incubated for an additional 4 h. Then, equal volume of solubilizing solution (dimethyl sulfoxide 40%, SDS 10% and acetic acid 2%) was added to the cell culture to dissolve the formazan crystals and incubated for 10 min at room temperature. The absorbance rate of the cell

culture was detected at 570 nm by using a Microplate Reader (Bio-Rad, Hercules, CA, USA). Each experiment was performed as biological quadruplicate.

### Clonogenic assay
Melanoma cells (2,000 cells/well) were seeded into 6-well plates, and following cell attachment, they were treated with DMSO or NB360 as indicated. Then, the plates were incubated at 37°C with 5% $CO_2$, until cells formed colonies (12–15 days). Colonies were fixed with 75% methanol and stained with 0.5% crystal violet, then rinsed with PBS, dried, and counted using the ImageJ software.

### Plasmid construction and generation
For the generation of the sh-BACE2 plasmid, annealed oligonucleotides were designed according to the RNA Consortium's recommendation (http://www.broadinstitute.org/rnai/trc) targeting GCACTCCTACA TAGACACGTA in the coding region of BACE2 and were cloned into pLKO-Tet-On by Age1 and EcoR1 sites to produce pLKO-Tet-On-shBACE2. The construct was confirmed through DNA sequencing.

### IGR37 infection and selection
To generate lentiviral particles, HEK293T cells were seeded in 10-cm dishes and transfected with 10 μg of pLKO-Tet-On-shBACE2, 2.8 μg of ENV (VSV-G) (Addgene, USA), 5 μg of pMDL (gag&pol) (Addgene, USA), and 2.5 μg of REV (Addgene, USA) adding $CaCl_2$ (Carlo Erba, Italy). After 72 h, the viral supernatant was recovered and filtered on a 0.45 μm filter (Millipore, USA). IGR37 cells cultured in 6 wells-cell culture plate (Corning, USA) were infected adding 1 ml of filtered viral supernatant in the presence of polybrene (8 μg/ml; Millipore, USA) by centrifugation for 20 min at 650 g. The infected cells were selected with 500 μg/μl of Neomycin G418 (Life Technologies, USA) added to the culture medium.

### Single-cell cloning for iBACE2 KD IGR37
After selection, iBACE2 KD IGR37 cells were sorted by FACS Aria (Becton Dickinson), and a single cell was seeded in a 96-well plate containing 250 μl of DMEM supplemented with 10% FBS S.A., 1% L-Glu, and 500 μg/μl of Neomycin G418 + 250 μl of conditioned medium. Cells were incubated at 37°C and 5% $CO_2$ until 70% of confluency and then moved to a bigger plate, up to 100-mm dish. shRNA knockdown was induced by adding 1 μg/ml of doxycycline.

### Study approval

Informed consent was obtained from all study participants. Study approval was given by the Institutional Review Board of the Grande Ospedale Metropolitano Niguarda. All cases of melanoma cancer were pathologically confirmed.

## Data availability

Proteomic datasets produced in this study are available in the following databases:

- Proteomics Identification database: PeptideAtlas PASS01358 (http://www.peptideatlas.org/PASS/PASS01358)

Expanded View for this article is available online.

## Acknowledgments

We thank the IFOM Functional Proteomics group and the Proteomics facility for critical comments and suggestions. We thank Luca Azzolin and Stefano Piccolo for YAP antibody and for their comments on our work. We acknowledge the Imaging, Mass Spectrometry and the Histopathology units at IFOM for their precious work. We thank Giannino Del Sal for discussion. We acknowledge Giuseppe Ossolengo for technical advice for gel filtration. We thank Jeffry W. Kelly for providing PMEL amyloidogenic fragment plasmid. We thank Marco Cirò from IFOM for providing pLKO-TET-on plasmid. Francesco Farris is a PhD student within the European School of Molecular Medicine (SEMM). We also thank Dr. Ulf Neumann from Novartis for kindly providing NB360 according to MTA agreement. This work has been supported by AIRC IG 18607 and IG 14578 to Angela Bachi.

## Author contributions

Methods development, VM, UR, and GM; Validation and formal analysis, VM; Investigation, VM, GM, FF, UR, ST, CB, FP, AS, and FC; Resources, AB, EB, SM, and LL; Writing original draft, VM, GM, FF, and AB; Supervision, project administration, and funding acquisition, AB.

## Conflict of interest

The authors declare that no conflict of interest exists.

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
