## [Review Process File · EMBO Reports]

Amyloid aggregates accumulate in melanoma metastasis modulating YAP activity

Vittoria Matafora, Francesco Farris, Umberto Restuccia, Simone Tamburri, Giuseppe Martano, clara bernardelli, Andrea Sofia, Federica Pisati, Francesca Casagrande, Luca Lazzari, Silvia Marsoni, Emanuela Bonoldi, and Angela Bachi

DOI: [10.15252/embr.202050446](https://doi.org/10.15252/embr.202050446)

Corresponding author(s): Angela Bachi (angela.bachi@ifom.eu)

Review Timeline:

Submission Date:	17th Mar 20
Editorial Decision:	17th Mar 20
Revision Received:	3rd Jun 20
Editorial Decision:	25th Jun 20
Revision Received:	27th Jun 20
Accepted:	7th Jul 20

Editor: Achim Breiling

Transaction Report: This manuscript was transferred to EMBO reports following peer review at The EMBO Journal.

Dear Dr. Bachi,

Thank you for transferring your manuscript to EMBO reports. I now went through your manuscript and the referee reports from The EMBO Journal (attached again below). All referees acknowledge the potential interest of the findings. Nevertheless, they have raised a number of concerns and suggestions to improve the manuscript, or to strengthen the data and the conclusions drawn.

EMBO reports emphasizes novel functional over detailed mechanistic insight, but asks for strong in vivo relevance of the findings, and clear experimental support of the major conclusions. Thus, we will not require to address points regarding more refined mechanistic details, but it would be important to provide some proof of physiological relevance, as indicated by referee #2 in his second point. Also all experimental inconsistencies indicated by the referees need to be addressed.

Given the constructive referee comments, we would like to invite you to revise your manuscript for EMBO reports with the understanding that the referee concerns must be addressed in the revised manuscript (as indicated above) and/or in a detailed point-by-point response. Acceptance of your manuscript will depend on a positive outcome of a second round of review (using the same referees that have assessed the study before). It is our policy to allow a single round of revision only and acceptance or rejection of the manuscript will therefore depend on the completeness of your responses included in the next, final version of the manuscript.

Revised manuscripts should be submitted within three months of a request for revision. We are aware that many laboratories cannot function at full efficiency during the current COVID-19/SARS-CoV-2 pandemic and we have therefore extended our 'scooping protection policy' to cover the period required for full revision. Please contact me to discuss the revision should you need additional time, and also if you see a paper with related content published elsewhere.

1) a .docx formatted version of the final manuscript text (including legends for main figures, EV figures and tables), but without the figures included. Please make sure that the changes are highlighted to be clearly visible. Figure legends should be compiled at the end of the manuscript text.

Please order the manuscript sections like this:

Title page - Abstract - Introduction - Results - Discussion - Materials and Methods - DAS - Acknowledgements - Author contributions - Conflict of interest - References - Figure legends - Expanded View Figure legends

2) individual production quality figure files as .eps, .tif, .jpg (one file per figure), of main figures and EV figures. Please upload these as separate, individual files upon re-submission.

For more details please refer to our guide to authors:

See also our guide for figure preparation:

http://wol-prod-cdn.literatumonline.com/pb-assets/embosite/EMBOPress_Figure_Guidelines_061115-1561436025777.pdf

4) a complete author checklist, which you can download from our author guidelines (<https://www.embopress.org/page/journal/14693178/authorguide>). Please insert page numbers in the checklist to indicate where the requested information can be found in the manuscript. The completed author checklist will also be part of the RPF.

Please also follow our guidelines for the use of living organisms, and the respective reporting guidelines: <http://www.embopress.org/page/journal/14693178/authorguide#livingorganisms>

5) that primary datasets produced in this study (e.g. RNA-seq, ChIP-seq and array data) are deposited in an appropriate public database.

The accession numbers and database should be listed in a formal "Data Availability" section (DAS - placed after Materials & Methods) that follows the model below. Please note that the Data Availability Section is restricted to new primary data that are part of this study. This is now mandatory (like the COI statement). If no primary datasets have been deposited in any database, please state this in this section (e.g. 'No primary datasets have been generated and deposited').

Data availability

6) We strongly encourage the publication of original source data with the aim of making primary data more accessible and transparent to the reader. The source data will be published in a separate source data file online along with the accepted manuscript and will be linked to the relevant figure. If you would like to use this opportunity, please submit the source data (for example scans of entire gels or blots, data points of graphs in an excel sheet, additional images, etc.) of your key experiments together with the revised manuscript. If you want to provide source data, please include size markers for scans of entire gels, label the scans with figure and panel number, and send one PDF file per figure.

8) Regarding data quantification and statistics, can you please specify, where applicable, the number "n" for how many independent experiments (biological replicates) were performed, the bars and error bars (e.g. SEM, SD) and the test used to calculate p-values in the respective figure legends. Please provide statistical testing where applicable, and also add a paragraph detailing this to the methods section. See:
<http://www.embopress.org/page/journal/14693178/authorguide#statisticalanalysis>

9) Please format the references according to our journal style. See:
<http://www.embopress.org/page/journal/14693178/authorguide#referencesformat>

Yours sincerely,

Achim Breiling
Editor
EMBO Reports

Referee #1:

In this manuscript titled "Amyloid aggregates accumulate in melanoma metastasis driving YAP

mediated tumor progression" by Metafora et al., authors propose a role of amyloid aggregates in modulating YAP activity in the context of metastatic melanoma. They compared the secretome of matched metastatic melanoma cell lines (IGR37 and WM266.4) vs. primitive tumor cell lines (IGR39 and WM115) and they found different proteins involved in amyloid metabolism (APOE, SORT1, QPCT and PMEL) specifically secreted by the metastatic cell lines. They independently validated this finding by using the Proteostat detection dye (i.e. a dye that stains for protein aggregates) and found an increase in both intracellular and extracellular staining. Moreover, these data were confirmed in other three paired (primitive vs. metastatic) melanoma cell lines. Further, they found increase in cholesterol metabolism and in PMEL expression in metastatic cell lines. The protein aggregates are also present in metastatic lesion of melanoma patients, validating their finding in vivo. Finally, the authors asked if the protein aggregates found in metastatic cells could play any role in metastatic behavior. They treated cells with NB360, a BACE1/2 inhibitor impairing PMEL function, they found decrease in protein aggregates (by proteostat detection dye) and a concomitant decrease in secreted AGRIN protein. Given that AGRIN was recently shown to regulate YAP, they then looked at YAP activity and found that NB360 (i.e. less amyloid aggregates, less AGRIN) causes a decrease YAP nuclear localization, decrease in secreted YAP targets and decrease in mRNA expression of a YAP target gene (CTGF). To demonstrate a relevant role of amyloid proteins on YAP function, they challenged primitive tumor cell line (IGR39) with conditioned medium of metastatic cell line (IGR37) and found increase in CTGF mRNA expression. Also, by adding recombinant PMEL amyloid fibrils to primitive tumor cell line (IGR39) they found increase in CTGF mRNA expression. Finally, combined treatment of NB360 and doxorubicin (a chemotherapeutic drug) caused an increase in chemotherapeutic sensitivity of metastatic cell line, by comparing cell viability.

The paper has two main messages: 1) metastatic melanoma cells enhance the production/secretion of amyloid aggregates compared to the primary tumor; 2) enhanced secretion of protein aggregates, containing amyloid fibrils together with other proteins (such as AGRIN), supports YAP activity.

The manuscript in its current form has multiple weaknesses currently precluding publication:

- A) there is no clue as to what regulates amyloid fibril production/secretion specifically in metastatic cells. This process is a natural process in melanocytes / pigment cells, so it remains unclear whether increased amyloid fibrils relate to differential "differentiation" of primary vs metastatic cells
- B) there is no clue as to whether this pathway is functionally relevant for metastatic cell migration/proliferation/survival/metastasis etc.
- C) the link with YAP remains circumstantial at best, without any functional consequence, and anyway obscure in its mechanisms (is YAP regulated by AGRIN or by fibrils? why the authors bring about ECM stiffness if they are working on plastics, which is several orders of magnitude stiffer than any physiological ECM?) and very weak experimentally (the authors compare cells treated with a conditioned medium from metastatic cells with no conditioned medium, which is a very poor control)
- D) there are some experimental inconsistencies, for example the authors show that the Bace inhibitor equally inhibits proliferation in primitive and metastatic cell lines, which is at odds with the finding that only metastatic cells secrete aggregates. Please note that this in principle detracts from a relevant role of secreted aggregates, and rather indicates a general role of intracellular PMEL.
- E) The authors claim that doxorubicin treatment works synergistically with Bace inhibitors, and that this is stronger in metastatic cells. However, in Fig. Supp. Fig. 12A-D, the data show that, in general, doxorubicin is more active on metastatic cells, and that Bace inhibitors slightly enhance these effects in both primitive and metastatic cells. Please note these data lack of any statistical analysis to support this claim.

Referee #2:

This is an interesting story raising the possibility that inhibitors of the protease BACE2 may be helpful for melanoma treatment. Specifically, the authors found secretome differences between metastatic and primitive melanoma cell lines (including ApoE) suggesting that protein aggregation, including of PMEL, may contribute to the metastatic phenotype. Because PMEL aggregation requires the protease BACE2 the authors zoom in on this protease and show that a BACE2-targeted inhibitor reduces cell proliferation in vitro and enhances susceptibility of the cell lines to cytotoxic drugs.

This study is novel and the proposed concept is exciting as it brings different fields together (melanoma, protein aggregation, Alzheimer's disease (through BACE2)). However, the present form of the manuscript has several major issues. First, it lacks evidence for physiological relevance of the proposed concept. Second, the BACE2 inhibitor experiments are not yet well controlled. Third, the authors pick - out of the many changed proteins - the ones that fit their hypothesis without some essential controls.

Major points:

1. Small molecule inhibitors often have off-target effects in addition to their intended effects. Thus, the authors need to repeat the BACE2-inhibitor experiment with a genetic approach, such as the use of siRNAs or lentiviral shRNA or even CRISPR against BACE2. The same approach needs to be done targeting BACE1, which should not give the same phenotype if the proposed concept with a key role for BACE2 is true. The authors should additionally use a second BACE2 inhibitor. Different ones are commercially available.
2. Physiological relevance. All experiments are done with cultured cell lines. The authors need to repeat a key experiment, in particular the BACE2 inhibitor/siRNA treatment, with primary (freshly isolated) melanoma cells or with an ex vivo system (if available) or best with a mouse model of melanoma.
3. Some experiments are only shown for one cell type (e.g. IGR) but not the other one or vice versa. One example is figure 3A, C, D. These are crucial experiments, which need to be validated in the other cell line as well. Otherwise, this suggests that only data on the one cell line are shown that did work, while the other cell line did not work for this experiment so that data were excluded.
4. What is the nature of the aggregates? The authors talk about aggregates, without providing evidence whether they mostly consist of PMEL or also of other aggregates and whether the PMEL aggregates are the most relevant ones. You may start by identifying the nature of the proteins by isolating the aggregates and doing mass spec. A crucial control experiment is then to knock-down/-out PMEL and demonstrating that this rescues the phenotype. Likewise, in the medium transfer experiment the authors may deplete PMEL aggregates (by immunoprecipitation) and then test rescue.
5. Fig.6: do a dose-response curve of the BACE inhibitor on cell proliferation. I guess that 25 uM will never be achievable in vivo. Thus, for the potential therapeutic applicability it will be important to see the effects (or part thereof) also at lower concentrations.

Minor points:

1. Figure 2: this is an unusual presentation of proteomic data. Please show additionally (at least in the supplement) a typical volcano plot, where Log₂ ratios are plotted against the -log₁₀ of the p-value (if enough replicates were done).
2. Describe in more detail how the proteostat aggregates are quantified and indicate whether the panels (e.g. Fig. 2F) were only used for quantifying intracellular or also extracellular aggregates.

Looking at panel 2G I am wondering whether the pictures in 2F are representative, because basically no aggregates are seen in the IGR-39, while they are clearly detected in 2G. Provide a representative picture (also for the other figures, such as Fig. 5D).

3. Be specific with the nomenclature of BACE1 versus BACE2. You write mostly about BACE, but the proteins are called BACE1 or BACE2. If you know which one you talk about, mention that one. If you are not sure about the identity, use something like BACE1/2 or the like. In contrast, for inhibitors that target both BACE1 and BACE2, it is common to refer to them as BACE inhibitors (without the 1 or 2).

4. Suppl. fig. 4B: does this volcano contain pooled data from all tested cell lines? Describe in more detail.

5. The authors claim on page 11 that NB-360 impairs maturation of APP. What exactly do they mean with this? Altered glycosylation?

6. Describe in more detail how YAP nuclear translocation was quantified.

Referee #3:

The authors have compared the secretome profiles of primary and metastatic melanoma cell lines using LC-MS proteomics to discover amyloid protein aggregation in metastatic melanoma cell lines. The proteomic data is extensive and appears well done. Moving from cell lines to patient samples, it is particularly striking to see protein aggregates in metastatic but not primitive patient tumors. I believe this study would be of interest to the melanoma and tumor microenvironment communities. However, I feel that there are critical gaps in this manuscript which prohibit publication at this time. In particular, the authors have demonstrated that "amyloid aggregates accumulate in melanoma metastasis" but they have not shown that amyloid aggregates drive "YAP mediated tumor progression" as claimed by their title. That is, there is no data showing that inhibition of YAP or amyloid protein secretion would prevent either tumor progression or metastasis. However, the finding the BACE inhibitors may represent a new therapeutic approach for melanoma would be highly significant.

The authors should also consider their data in light of another pre-print that appears to show complementary data: Melanoma-secreted Amyloid Beta Suppresses Neuroinflammation and Promotes Brain Metastasis, <https://www.biorxiv.org/content/10.1101/854885v1>.

Major concerns:

- The authors have not presented any data showing that amyloid aggregates drive tumor progression and/or metastasis. They have shown that metastatic cell lines and patient tumors accumulate amyloids, and they have perhaps shown that metastatic cell lines are more sensitive to BACE inhibition (see comment below about drug synergy). But if the authors want to claim that amyloid and/or YAP drives tumor progression, they need to provide stronger data to support this claim.

- The mechanistic data connecting extracellular secreted amyloid proteins to YAP is weak, as the authors have shown only that YAP nuclear translocation is reduced by the BACE inhibitor. In fact, the authors have chosen to highlight YAP in their title, but only one panel of one figure (5F) has data related to YAP. Because YAP nuclear translocation is restricted by serine phosphorylation at multiple sites (e.g., Ser127) (PMID: 17974916 & 22863277), the authors should test whether YAP phosphorylation is increased by BACE inhibition and reduced by stimulation with IGR37 conditioned media and PMEL fibril stimulation.

- In addition, the authors have used only one inhibitor and no genetic manipulations to demonstrate

the involvement of BACE in amyloid protein secretion. The authors should include a genetic experiment to prove that BACE activity is required for amyloid protein aggregation.

- Figs. 2B, C: The authors should present this data as a traditional volcano plot where the y-axis is -log(p-value) not log(intensity). On the current plots, the reader cannot assess how statistically different the protein levels are between the primitive/metastatic tumor cells.
- Tables S1-S5 with secretome data should include *ALL* proteins identified with the intensities measured in all biological and technical replicates. Including such data in an easily accessible format will improve future re-analysis of the data by other researchers.
- Page 14, Figure 6: How can the authors claim that NB360 and doxorubicin are synergistic in IGR37 but not IGR39 when they have not measured the IC50 for NB360? It would be better if the authors could measure an accepted measurement of synergy such as Combination Index (PMID: 20068163)
- Figure 6: what is the concentration of NB360 used in panels C, D, and E? It is not described in the legend or figure.
- Can they measure changes in stiffness?
- genetic modulation of amyloid secretion? Agrin knockdown?

Minor concerns:

- Grammar needs editing of a fluent English speaker. There are numerous grammatical errors in the abstract alone.
- Page 5: authors state that the metastatic cell lines have "increased ability to undergo unlimited division". This is not supported by the data. Aren't both of these cell lines immortalized? Thus, they would have the same capacity for "unlimited division". I think the authors are referring to the capacity for RAPID division.
- Fig 2: volcano plots are fuzzy (low image resolution) in the pdf copy that I have. This should be improved so that readers can see the protein names more clearly.
- Why have the authors added the Proteostat data for the WM cell lines to Supp. Fig. 1 after the discussion of Supp. Figs. 2 and 3? For better flow of the Supp. Figs, it would be better to move this data below Supp. Figs. 2 and 3, perhaps as a new Supp. Fig. 4.
- The volcano plots in Supp. Fig. 5 have metastatic cells on the left. The plots in Fig. 2 have metastatic on the right. The authors should standardize so that metastatic cells are always on the same side of the volcano plots.
- Page 18: The authors state "Dr Richard Hynes, from MIT, recently demonstrated in a PNAS paper that in vivo ECM production is mostly fibroblastic, while ECM remodeling is both tumor cell and fibroblastic cell dependent." This appears to be a reference to Lamar et al (2012). If so, the authors should correctly reference the paper. Also, the reference to MIT should be removed because this is not relevant to the results of the aforementioned paper.

Referee #1:

In this manuscript titled "Amyloid aggregates accumulate in melanoma metastasis driving YAP mediated tumor progression" by Matafora et al., authors propose a role of amyloid aggregates in modulating YAP activity in the context of metastatic melanoma. They compared the secretome of matched metastatic melanoma cell lines (IGR37 and WM266.4) vs. primitive tumor cell lines (IGR39 and WM115) and they found different proteins involved in amyloid metabolism (APOE, SORT1, QPCT and PMEL) specifically secreted by the metastatic cell lines. They independently validated this finding by using the Proteostat detection dye (i.e. a dye that stains for protein aggregates) and found an increase in both intracellular and extracellular staining. Moreover, these data were confirmed in other three paired (primitive vs. metastatic) melanoma cell lines. Further, they found increase in oxysterol metabolism and in PMEL expression in metastatic cell lines. The protein aggregates are also present in metastatic lesion of melanoma patients, validating their finding in vivo. Finally, the authors asked if the protein aggregates found in metastatic cells could play any role in metastatic behavior. They treated cells with NB360, a BACE1/2 inhibitor impairing PMEL function, they found decrease in protein aggregates (by proteostat detection dye) and a concomitant decrease in secreted AGRIN protein. Given that AGRIN was recently shown to regulate YAP, they then looked at YAP activity and found that NB360 (i.e. less amyloid aggregates, less AGRIN) causes a decrease YAP nuclear localization, decrease in secreted YAP targets and decrease in mRNA expression of a YAP target gene (CTGF). To demonstrate a relevant role of amyloid proteins on YAP function, they challenged primitive tumor cell line (IGR39) with conditioned medium of metastatic cell line (IGR37) and found increase in CTGF mRNA expression. Also, by adding recombinant PMEL amyloid fibrils to primitive tumor cell line (IGR39) they found increase in CTGF mRNA expression. Finally, combined treatment of NB360 and doxorubicin (a chemotherapeutic drug) caused an increase in chemotherapeutic sensitivity of metastatic cell line, by comparing cell viability.

The paper has two main messages: 1) metastatic melanoma cells enhance the production/secretion of amyloid aggregates compared to the primary tumor; 2) enhanced secretion of protein aggregates, containing amyloid fibrils together with other proteins (such as AGRIN), supports YAP activity.

The manuscript in its current form has multiple weaknesses currently precluding publication:

A) there is no clue as to what regulates amyloid fibril production/secretion specifically in metastatic cells. This process is a natural process in melanocytes / pigment cells, so it remain unclear whether increased amyloid fibrils relate to differential "differentiation" of primary vs metastatic cells

We thank the reviewer for the observation, it is true that this process occurs also in normal melanocyte where PMEL amyloid fibrils are retained into melanosomes (van Niel et al., 2015, Cell Reports). What we show here by proteostat staining of normal skin biopsies is no positivity for the dye indicating undetectability of protein aggregates in the normal skin and primary tumor, while the presence of protein aggregates is associated with the metastatic phenotype both in cell lines and in human biopsies (fig 4 and fig EV3, panel C).

In the cellular model we demonstrated that aggregates are present in the secretome of metastatic cells and we proved that the inhibition of BACE2 (overexpressed in metastasis vs primary tumor cells) reduces the presence of protein aggregates. Further, TCGA data confirm that not only BACE2 but also other proteins involved in amyloidogenesis, such as APOE, are upregulated in tumor compared to normal tissue. Overall, we demonstrated that protein aggregates differentiate metastatic melanoma from primitive melanoma, that BACE2 is more expressed in metastatic melanoma compared to primitive melanoma and that inhibition of BACE2 decreases the level of protein aggregates in metastatic cells. At this stage we do not

know what is regulating the different expression of BACE2 and if this is sufficient to explain a differential "differentiation".

We have collected some interesting data showing an increased level of MITF (microphthalmia-associated transcription factor) in metastatic cells vs primary ones (figure below).

Additionally, we have interesting results that point to a role of fibrils in the differentiation from primitive to metastatic cells as amyloid fibrils added to primitive cells are able to promote MITF activation (figure below). We might therefore suppose that amyloids fibrils are associated or are involved into a "differentiation" of metastatic vs primary cells. Further experiments are required to confirm this hypothesis therefore we did not include these data in the manuscript.

B) there is no clue as to whether this pathway is functionally relevant for metastatic cell migration/proliferation/survival/metastasis etc.

Our study show that the pathway related to amyloid fibrils production/secretion is specific of metastatic cells. As shown in Fig. 1B and Fig. EV1A, metastatic cell lines show higher proliferation rate compared to the matched primitive cell line, which in turn has higher mobility when monitored alive using time-lapse microscopy (Fig. 1D, Supplementary Videos 1-2). We have also shown that amyloid fibrils potentially have a role in the metastatic tumor growth as we demonstrated that, by impairing amyloids production/secretion (inhibiting BACE2 or in BACE KD cells), metastatic cells proliferate less (Fig 6 and Fig EV5A,B).

Regarding cell survival, we showed that both BACE 2 KD and BACE inhibition are not cytotoxic (Fig EV3), therefore we suppose that protein aggregates affect proliferation but not cell survival. Supporting this hypothesis, we do not observe any increase of the apoptotic marker Annexin V (fig below).

Moreover, BACE1/2 inhibition increases susceptibility to drugs especially in metastatic cell lines attesting that amyloids fibrils might be functionally relevant in these cells (Fig 6).

C) the link with YAP remains circumstantial at best, without any functional consequence, and anyway obscure in its mechanisms (is YAP regulated by AGRIN or by fibrils? why the authors bring about ECM stiffness if they are working on plastics, which is several orders of magnitude stiffer than any physiological ECM?) and very weak experimentally (the authors compare cells treated with a conditioned medium from metastatic cells with no conditioned medium, which is a very poor control)

YAP is recognized as nuclear sensors of mechanical clues in response to extracellular matrix (ECM) signals. Recently, it has been proposed that Agrin acts as a mechanotransduction signal in the matrix. Agrin transduces cellular rigidity signals that prompt YAP activation through the Lrp4 receptor mediated signaling pathways. (Chakarboty et al. 2015). Moreover, ECM stiffness enhanced Agrin levels and increased FAK activation which drives YAP nuclear translocation (Zebda N. et al. 2012, Chakarboty et al. 2015, Chakarboty et al. 2017). In the present work we have demonstrated that amyloids fibrils in metastatic melanoma induce YAP nuclear localization and its transcriptional activity. In the revised manuscript we added novel data that demonstrate that PMEL fibrils are also able to enhance Agrin levels and increase FAK activation (Fig EV4 L,M). These findings suggest that amyloid fibrils might trigger the Agrin-pFAK-YAP axis.

We are conscious that working on plastic we cannot measure stiffness and we know that further analysis is needed to confirm that fibrils are active components of the ECM stiffness. We agree with the reviewer that we cannot discuss about ECM stiffness as we have only identified protein aggregates and we do not know at this stage if they affect extracellular stiffness. We have changed the manuscript accordingly.

For the experiment with the conditioned medium, as reported in the methods session, we used as control a conditioned medium from IGR39 cells cultured in the same condition as IGR37 cells.

D) there are some experimental inconsistencies, for example the authors show that the Bace inhibitor equally inhibits proliferation in primitive and metastatic cell lines, which is at odds with the finding that only metastatic cells secrete aggregates. Please note that this in principle detracts from a relevant role of secreted aggregates, and rather indicates a general role of intracellular PMEL.

We evaluated the clonogenic activity of IGRs and WMs melanoma cells upon treatment with NB360, and we found a diminished formation of new colonies (Fig EV5) and a decreased proliferation rate (Fig. 6A) in primitive and metastatic cell lines. We agree that these data

might be in contrast with the finding that only metastatic cells secrete aggregates. However, we have to consider that BACE1 and BACE2 might regulate different proteins in primitive and metastatic cells which might impair cell proliferation by different pathways. To address this issue, we selected the specific BACE2 inhibitor 3I (A.R. Ghosh et al., 2019, ChemMedChem) that is the compound that shows the highest difference in term of Ki between BACE2 and BACE1 (BACE2 Ki = 1,6nM; BACE1 Ki = 815,1 nM). We treated primary and metastatic cells and, as it is reported in Fig 6b in the revised manuscript, we showed that only metastatic cells are affected by 3I demonstrating a link between BACE2 and proliferation in metastasis but not in primitive melanoma cells.

E) The authors claim that doxorubicin treatment works synergistically with Bace inhibitors, and that this is stronger in metastatic cells. However, in Fig. Supp. Fig. 12A-D, the data show that, in general, doxorubicin is more active on metastatic cells, and that Bace inhibitors slightly enhance these effects in both primitive and metastatic cells. Please note these data lack of any statistical analysis to support this claim.

In order to assess if BACE inhibitor potentiates doxorubicin mediated toxicity, MTT assay was performed to determine the IC50 of doxorubicin in presence of NB360 (Fig. 6C-D) in primitive and metastatic cells. MTT assay was performed in IGR39 and IGR37 cells treated with different dose of doxorubicin in absence or presence of 25 μ M of NB360. The IC50 for doxorubicin alone, is not different between primary and metastatic cells (Fig 6E), instead, when used in combination with NB360, the doxorubicin IC50 is significantly higher (Ttest, pvalue <0.05 N=4) in primitive cells compared to metastatic cells (fig 6E), suggesting that the presence of amyloid fibrils might contribute to an increased susceptibility to the chemotherapeutic agent. We tested if the combo performed equally in WMs melanoma cell lines. We verified the strongest condition in which the effect of the combo was higher and we confirmed that the combo is more effective compared to doxo alone (Fig EV5D). Condition: doxorubicin 10 μ M, NB-360 25 μ M, T-test, pvalue<0.05).

Referee #2:

This is an interesting story raising the possibility that inhibitors of the protease BACE2 may be helpful for melanoma treatment. Specifically, the authors found secretome differences between metastatic and primitive melanoma cell lines (including ApoE) suggesting that protein aggregation, including of PMEL, may contribute to the metastatic phenotype. Because PMEL aggregation requires the protease BACE2 the authors zoom in on this protease and show that a BACE2-targeted inhibitor reduces cell proliferation in vitro and enhances susceptibility of the cell lines to cytotoxic drugs.

This study is novel and the proposed concept is exciting as it brings different fields together (melanoma, protein aggregation, Alzheimer's disease (through BACE2)). However, the present form of the manuscript has several major issues. First, it lacks evidence for physiological relevance of the proposed concept. Second, the BACE2 inhibitor experiments are not yet well controlled. Third, the authors pick - out of the many changed proteins - the ones that fit their hypothesis without some essential controls.

Major points:

1. Small molecule inhibitors often have off-target effects in addition to their intended effects. Thus,

the authors need to repeat the BACE2-inhibitor experiment with a genetic approach, such as the use of siRNAs or lentiviral shRNA or even CRISPR against BACE2. The same approach needs to be done targeting BACE1, which should not give the same phenotype if the proposed concept with a key role for BACE2 is true. The authors should additionally use a second BACE2 inhibitor. Different ones are commercially available.

We thank the reviewer for recognizing the novelty of our work and for the useful suggestions that we have followed in order to improve the quality of our work. In particular, to eliminate possible off-target effects, we have reproduced the BACE inhibitor experiment with a genetic approach. We have used Lentiviral vector-mediated doxycycline-inducible BACE2 shRNA. We have confirmed that BACE2 regulates PMEL amyloid secretion in metastatic melanoma IGR37 cells (Fig EV4D) and we have also confirmed that the YAP target CTGF is downregulated as upon NB360 treatment (fig EV4G).

We have also selected a specific BACE2 inhibitor (3I, BACE2 Ki = 1,6nM; BACE1 Ki = 815,1 nM; A.R. Ghosh et al., 2019, ChemMedChem) that abrogates protein aggregates formation (Fig EV4H) and affects YAP transcriptional activity as measured by CTGF expression (Fig EV4I). We have added these data in the revised version of the manuscript.

Therefore, we can conclude that indeed PMEL amyloid-like structures are processed by BACE2 and are able to induce YAP activation in metastatic melanoma cells.

2. Physiological relevance. All experiments are done with cultured cell lines. The authors need to repeat a key experiment, in particular the BACE2 inhibitor/siRNA treatment, with primary (freshly isolated) melanoma cells or with an ex vivo system (if available) or best with a mouse model of melanoma.

The translation of our findings in vivo is clearly the long term aim of our work. For this reason, we are already planning and looking for collaborators as well as for ministerial permission to have access to fresh human material and animal models.

Unfortunately, this is not achievable in the timeframe of a revision but will for sure be the subject of a future follow up of this work.

Moreover, very recent publications (Kleffman, K. et al., BioRxiv 2019 and Ostendorf, B. N. et al., Nature medicine 2020) point out the relevance of the pathway that we are here describing for the first time in its molecular details.

3. Some experiments are only shown for one cell type (e.g. IGR) but not the other one or vice versa. One example is figure 3A, C, D. These are crucial experiments, which need to be validated in the other cell line as well. Otherwise, this suggests that only data on the one cell line are shown that did work, while the other cell line did not work for this experiment so that data were excluded.

We disagree with this comment as we have observed an increased secretion of amyloid like aggregates and proteins that assist amyloid maturation in IGR metastatic compared to IGR primitive, in WM metastatic vs WM primitive cells (figure 2 B-E) and also in a cohort of other three metastatic vs three primitive cell lines that are MEWO, SK-MEL-5 and SK-MEL-28 vs A-375, C32 and IPC-298 (fig EV2) and in human biopsies (figure 4).

Oxysterol measurements are consistent in WM cells (Figure 3 A) and in SK-MEL-5, SK-MEL-28, A-375, IPC-298 as shown in figure 3B.

Data on PMEL expression have been verified in all analysed cell lines (see proteomics data Dataset EV1-8).

We have confirmed that BACE inhibition affects proliferation in both IGRs and WMs cell lines and that the combo of doxorubicin and BACE inhibitor works both on IGRs and WMs cell lines (figure 6 A,B and figure EV5A,C).

Therefore, we think that crucial experiments have been confirmed in more than one cell line attesting the robustness of the presented data.

4. What is the nature of the aggregates? The authors talk about aggregates, without providing evidence whether they mostly consist of PMEL or also of other aggregates and whether the PMEL aggregates are the most relevant ones. You may start by identifying the nature of the proteins by isolating the aggregates and doing mass spec. A crucial control experiment is then to knock-down/-out PMEL and demonstrating that this rescues the phenotype. Likewise, in the medium transfer experiment the authors may deplete PMEL aggregates (by immunoprecipitation) and then test rescue.

Actually, we do not know yet the nature of the aggregates. From the analysis of the metastatic secretome, we found that there are several proteins with amyloidogenic properties as PMEL, APP, APLP1, APLP2. As we also found the amyloidogenic machinery active (APOE, QPCT, SORT1) we can suppose that the detected fibrils might derive from all these amyloidogenic proteins. We followed PMEL as it is the most expressed and conserved among all the metastatic cells analyzed. We agree that further experiments might be useful to elucidate the nature of the aggregates that we found, and we are planning them. On the other hand, we also demonstrated that PMEL fibrils alone are able to recapitulate and rescue the phenotype that we associated to the presence of protein aggregates.

5. Fig.6: do a dose-response curve of the BACE inhibitor on cell proliferation. I guess that 25 uM will never be achievable in vivo. Thus, for the potential therapeutic applicability it will be important to see the effects (or part thereof) also at lower concentrations.

In agreement with the reviewer, we followed a dose dependent analysis of cell viability and we found that the dose we used is not cytotoxic (fig EV3H).

Minor points:

1. Figure 2: this is an unusual presentation of proteomic data. Please show additionally (at least in the supplement) a typical volcano plot, where Log₂ ratios are plotted against the -log₁₀ of the p-value (if enough replicates were done).

In figure 2 we showed scatter plot visualization of proteomic data. Scatter plots visualize heavy/light ratios and the amount of the proteins (intensity). p-values are provided by the color of the dots. Exactly, red dots represent proteins that were significant with FDR <0.05, blue dots represents proteins with p-value<0.05. p-value is calculated using the Perseus software provided in the MaxQuant package. As requested, we report below a typical volcano plot, where Log₂ ratios are plotted against the -log₁₀ of the p-value.

Volcano plot for metastatic vs primitive melanoma cell lines secretome (<http://volcanor.bioinf.su>).

- A.** Volcano plot of SILAC ratios for the secretome of metastatic IGR37 vs primitive IGR39 cells.
- B.** Volcano plot of SILAC ratios for the secretome of metastatic WM266.4 vs primitive WM115 cells.

2. Describe in more detail how the proteostat aggregates are quantified and indicate whether the panels (e.g. Fig. 2F) were only used for quantifying intracellular or also extracellular aggregates. Looking at panel 2G I am wondering whether the pictures in 2F are representative, because basically no aggregates are seen in the IGR-39, while they are clearly detected in 2G. Provide a representative picture (also for the other figures, such as Fig. 5D).

We apologize for the lack of details. For intracellular analysis of protein aggregates, confocal fluorescence images of Proteostat (1:2000, red spots) and DAPI staining (blue) were acquired and all the dots and nuclei were counted by using the counter in Fiji software. The number of

aggregates/cell of at least three biological replicates was used for the statistical analysis; we took into account all types of aggregates the small and the bigger one. For extracellular analysis of protein aggregates, fluorescence gain of secreted proteins treated with Proteostat reagent was used for the quantitative analysis. Proteostat 1:1000 was added to secreted proteins. Fluorescence generated signals were read using an excitation setting of about 550 nm and an emission filter of about 600 nm.

3. Be specific with the nomenclature of BACE1 versus BACE2. You write mostly about BACE, but the proteins are called BACE1 or BACE2. If you know which one you talk about, mention that one. If you are not sure about the identity, use something like BACE1/2 or the like. In contrast, for inhibitors that target both BACE1 and BACE2, it is common to refer to them as BACE inhibitors (without the 1 or 2).

We thank the reviewer and we have modified the nomenclature as suggested in the revised manuscript.

4. Suppl. fig. 4B: does this volcano contain pooled data from all tested cell lines? Describe in more detail.

Volcano plot is calculated on LFQ intensities of all cell lines separately. We modified the legend of Fig EV2C,D. The LFQ intensities from all cell lines were uploaded in Perseus software for the construction of the volcano plot.

5. The authors claim on page 11 that NB-360 impairs maturation of APP. What exactly do they mean with this? Altered glycosylation?

When we write ‘maturation’, we consider the maturation into fibrils and not PTMs.

6. Describe in more detail how YAP nuclear translocation was quantified.

YAP nuclear translocation was quantified by dividing the confocal analyzed cells into three groups. One in which YAP signal is spread along the cell, one in which the signal is excluded from the nucleus and one in which the signal is exclusively in the nucleus. We counted the number of the cells of the three groups manually using the counter in Fiji. We reported the number of cells quantified in the three groups for three biological replicates.

Referee #3:

Manuscript #EMBOJ-2020-104662

Amyloid aggregates accumulate in melanoma metastasis driving YAP mediated tumor progression

Matafora et al

The authors have compared the secretome profiles of primary and metastatic melanoma cell lines using LC-MS proteomics to discover amyloid protein aggregation in metastatic melanoma cell lines. The proteomic data is extensive and appears well done. Moving from cell lines to patient samples, it is particularly striking to see protein aggregates in metastatic but not primitive patient

tumors. I believe this study would be of interest to the melanoma and tumor microenvironment communities. However, I feel that there are critical gaps in this manuscript which prohibit publication at this time. In particular, the authors have demonstrated that "amyloid aggregates accumulate in melanoma metastasis" but they have not shown that amyloid aggregates drive "YAP mediated tumor progression" as claimed by their title. That is, there is no data showing that inhibition of YAP or amyloid protein secretion would prevent either tumor progression or metastasis. However, the finding the BACE inhibitors may represent a new therapeutic approach for melanoma would be highly significant.

The authors should also consider their data in light of another pre-print that appears to show complementary data: Melanoma-secreted Amyloid Beta Suppresses Neuroinflammation and Promotes Brain Metastasis, <https://www.biorxiv.org/content/10.1101/854885v1>.

Major concerns:

1) The authors have not presented any data showing that amyloid aggregates drive tumor progression and/or metastasis. They have shown that metastatic cell lines and patient tumors accumulate amyloids, and they have perhaps shown that metastatic cell lines are more sensitive to BACE inhibition (see comment below about drug synergy). But if the authors want to claim that amyloid and/or YAP drives tumor progression, they need to provide stronger data to support this claim.

We agree with the reviewer that we have not demonstrated that YAP or aggregates impairment affect tumor progression therefore, the title has been modified focusing better on our results.

2) The mechanistic data connecting extracellular secreted amyloid proteins to YAP is weak, as the authors have shown only that YAP nuclear translocation is reduced by the BACE inhibitor. In fact, the authors have chosen to highlight YAP in their title, but only one panel of one figure (5F) has data related to YAP. Because YAP nuclear translocation is restricted by serine phosphorylation at multiple sites (e.g., Ser127) (PMID: 17974916 & 22863277), the authors should test whether YAP phosphorylation is increased by BACE inhibition and reduced by stimulation with IGR37 conditioned media and PMEL fibril stimulation.

We did not measure the level of YAP phosphorylation but we demonstrated through canonical experiments the activity of YAP (nuclear localization by Immunofluorescence and qPCR on YAP target genes) upon BACE inhibition (Fig.5F-G), stimulation with IGR37 conditioned media (Fig.5H) and PMEL fibrils stimulation (Fig.5I).

Additionally, we have obtained new data (fig EV4 I-L) showing that PMEL fibrils are able not only to promote CTGF expression but also to increase the expression of Agrin and FAK phosphorylation that is recognized as a mark of activated mechanotransduction (Zebda N. et al. 2012, Chakarboty et al. 2015, Chakarboty et al. 2017). It has been reported that Agrin sustains the mechano-responsiveness of YAP via FAK phosphorylation in a stiffness sensed manner which correlates with cell growth (Chakarboty S., Cell Rep. 2017). Moreover, Focal adhesion kinase (FAK) is recognized as a key molecule recruited in focal adhesions that respond to external mechanical stimuli. It has also been shown that FAK controls the nuclear translocation and activation of YAP in response to mechanical activation (Lachowski et al The Faseb journal 2018). For these reasons we think that our data strongly suggest that amyloid fibrils are able to induce mechanical signals that activate the Agrin-FAK-YAP axis.

3) In addition, the authors have used only one inhibitor and no genetic manipulations to demonstrate the involvement of BACE in amyloid protein secretion. The authors should include a genetic experiment to prove that BACE activity is required for amyloid protein aggregation.

We thank the reviewer for the suggestion and we have repeated the BACE inhibitor experiment with a genetic approach as suggested. In particular, we have used Lentiviral vector-mediated doxycycline-inducible BACE2 shRNA. We have confirmed that BACE2 regulates PMEL amyloid secretion in metastatic melanoma IGR37 (Fig EV4D) and we have also confirmed that the YAP target CTGF is downregulated as upon NB360 treatment (Fig EV4G). Therefore, we might conclude that indeed amyloid-like structures are processed by BACE2 and are able to induce YAP activation in metastatic melanoma cells. We added these new data in the revised manuscript.

4) Figs. 2B, C: The authors should present this data as a traditional volcano plot where the y-axis is $-\log(p\text{-value})$ not $\log(\text{intensity})$. On the current plots, the reader cannot assess how statistically different the protein levels are between the primitive/metastatic tumor cells.

Scatter plot visualization of proteomic data are commonly used to show SILAC ratios. The output of a SILAC workflow is usually the protein heavy/light ratios. Scatter plots visualize these ratios and the amount of the proteins (intensity). p-values are provided in the supplementary tables under the “significance” columns. Significance B is the p-value that is derived considering the Gaussian distribution of the ratios, it is calculated using the Perseus software provided in the MaxQuant package. Intuitively, the proteins that have higher ratios are the most significant therefore are in the extreme sides of the scatter plot. We have implemented this information in the plot by coloring the significant proteins.

5) Tables S1-S5 with secretome data should include *ALL* proteins identified with the intensities measured in all biological and technical replicates. Including such data in an easily accessible format will improve future re-analysis of the data by other researchers.

We apologize for the mistake, we have now updated all tables.

6) Page 14, Figure 6: How can the authors claim that NB360 and doxorubicin are synergistic in IGR37 but not IGR39 when they have not measured the IC50 for NB360? It would be better if the authors could measure an accepted measurement of synergy such as Combination Index (PMID: 20068163)

Following the guidelines proposed in the suggested paper we cannot claim that the combo NB360 and doxorubicin are synergistic as we do not have the dose dependent analysis of NB360 alone and we miss the IC50 for NB360.

On the other hand, as also the paper above suggests, we can conclude that the combined therapy of doxorubicin and NB360 is more efficient compared to doxorubicin alone, as $A+B > B$. (A=NB360, B=doxorubicin). We changed the text accordingly.

7) Figure 6: what is the concentration of NB360 used in panels C, D, and E? It is not described in the legend or figure.

The concentration of NB360 used in panels C, D, and E is 25uM. We added the concentration in the revised manuscript.

8) Can they measure changes in stiffness?

We agree that to further prove that fibrils induce mechanical signal which activate amyloid-agrin-YAP cascade we should measure the level of stiffness. We plan such experiments for the future follow up of our work. We have corrected the text accordingly.

9) genetic modulation of amyloid secretion? Agrin knockdown?

Up to now we only provided data with BACE 2 KD. In particular we have reproduced the BACE inhibitor experiment with a genetic approach. We have used Lentiviral vector-mediated doxycycline-inducible BACE2 shRNA. We have confirmed that BACE2 regulates PMEL amyloid secretion in metastatic melanoma IGR37 cells (Fig EV4D) and we have also confirmed that the YAP target CTGF is downregulated as upon NB360 treatment (fig EV4G).

Minor concerns:

- Grammar needs editing of a fluent English speaker. There are numerous grammatical errors in the abstract alone.
- Page 5: authors state that the metastatic cell lines have "increased ability to undergo unlimited division". This is not supported by the data. Aren't both of these cell lines immortalized? Thus, they would have the same capacity for "unlimited division". I think the authors are referring to the capacity for RAPID division.

We thank the reviewer for the observation, we have changed the text with rapid division.

- Fig 2: volcano plots are fuzzy (low image resolution) in the pdf copy that I have. This should be improved so that readers can see the protein names more clearly.

Images resolution has been improved.

- Why have the authors added the Proteostat data for the WM cell lines to Supp. Fig. 1 after the discussion of Supp. Figs. 2 and 3? For better flow of the Supp. Figs, it would be better to move this data below Supp. Figs. 2 and 3, perhaps as a new Supp. Fig. 4.

We agree with the reviewer, we have done a new Fig EV1 with proteostat data for the WM cells.

- The volcano plots in Supp. Fig. 5 have metastatic cells on the left. The plots in Fig. 2 have metastatic on the right. The authors should standardize so that metastatic cells are always on the same side of the volcano plots.

Thanks for the suggestion, we have now standardized all figures.

- Page 18: The authors state "Dr Richard Hynes, from MIT, recently demonstrated in a PNAS paper that in vivo ECM production is mostly fibroblastic, while ECM remodeling is both tumor cell and fibroblastic cell dependent." This appears to be a reference to Lamar et al (2012). If so, the authors should correctly reference the paper. Also, the reference to MIT should be removed because this is not relevant to the results of the aforementioned paper.

We have changed the reference, we apologize for the mistake.

Dear Dr. Bachi,

Thank you for the submission of your revised manuscript to our editorial offices. We have now received the reports from the three referees that were asked to re-evaluate your study, you will find below. As you will see, the referees #1 and #3 support the publication of your study in EMBO reports, whereas referee #2 has remaining concerns. Referee #3 has also some further issues and suggestions to improve the manuscript. Considering the constructive reports, we ask you to address these points in a final revised manuscript. Please also provide a point-by-point response addressing these remaining concerns.

- Please order the manuscript sections like this:

Title page - Abstract - Introduction - Results - Discussion - Materials and Methods - DAS - Acknowledgements - Author contributions - Conflict of interest - References - Figure legends - Expanded View Figure legends. The financial support part should be added to the acknowledgements. Please make sure that the granting information in the acknowledgements and in the manuscript submission system are complete and identical.

- Please provide the abstract written in present tense.

- Presently, there are no callouts to Fig. 4D, and the datasets EV 5,7 and 8. Please carefully check that all uploaded items are called out.

- Please provide legends for the movies. These should be separate text files, which are zipped together with the movie file and uploaded together. Do not add the legends to the main text file.

- Please make sure that the scale bars on the microscopic images are of equal thickness and well visible. Do not write on the scale bars. Please indicate their size in the respective figure legends.

- Finally, please find attached a word file of the manuscript text (provided by our publisher) with changes we ask you to include in your final manuscript text, and some queries, we ask you to address. Please provide your final manuscript file with track changes, in order that we can see any modifications done.

In addition I would need from you:

- a short, two-sentence summary of the manuscript
- two to three bullet points highlighting the key findings of your study

Kind regards,

Achim Breiling
Editor
EMBO Reports

Referee #1:

The authors have satisfied my major concerns.

Referee #2:

In reference to my previous points, the authors have not addressed enough my concerns to strengthen their claims. The manuscript remains very correlative, and in the end the experimental evidence does not support the proposed model.

point A: the observation of increased amyloid aggregates remains without any mechanism, apart from the overexpression of BACE2 in metastatic cells. I had proposed this as a possible manner to enrich the core message and to compensate weaknesses in other parts of the manuscript, but this was not expanded.

point B: the authors should be careful in writing (text and figures) "proliferation" when they just count the number of cells. The new data still do not provide any direct proof that the cell cycle is altered, for which a PI staining or EdU incorporation would easily provide support. Annexin V staining excludes death from classical apoptosis, but cell number could still change due to other types of death. Moreover, please be consistent on the Y axis of graphs: in figure 6, for example, A, B, C, D are all MTT assays, but this is labeled "cell number %" in A-B and "cell viability, % of control" in C-D which may lead the reader to believe these are different and more specific assays. What is quantified is just (indirectly) the number of cells.

point C: the data can support the notion that secreted PMEL fibrils activate YAP, provided that the authors can show that YAP localization in cells treated with NB360 is rescued by adding back IGR37 CM or PML fibrils.

Any reference to mechanotransduction is otherwise inappropriate. The sole correlation with FAK activity and Agrin secretion cannot be claimed to be "mechanotransduction" in absence of specific measures of the cytoskeleton, cell contractility etc., and without a clear demonstration for the FUNCTIONAL involvement of these players, starting from Agrin for which the data are only correlative. So, any mention to mechanotransduction should be avoided in the title, abstract and text. This can be of course discussed, in light of correlative evidence.

The same applies for Agrin: one thing is showing correlation with a secreted protein, another is providing experimental functional support that Agrin is REQUIRED downstream of fibrils, which is not provided here - and, thus, should not be claimed to be the mechanism by which YAP is being regulated.

Collectively, the link with YAP thus remains an observation without a mechanism, and without a functional relevance for the BACE/fibril phenotypes.

point D: the new data with the new BACE inhibitor are better in line with the proposed model. Still, at face value, all the rest of the functional evidence supporting the author's claims is based on the use of the least specific inhibitor, which may raise some specificity issues. Moreover, the functional role of the fibrils is almost uniquely based on the use of BACE inhibitors, which will also inhibit other pathways in cells beyond fibrils. So, to claim that the observed effects of BACE are through fibrils, the authors should provide key data throughout the paper (proliferation, YAP, chemotherapy etc) showing that an alternative means of inhibiting fibrils (e.g. knockdown of PMEL?) induces the same

phenotypes.

Referee #3:

The authors have reasonably addressed my previous points, but four more points need to be addressed:

Major points:

1. The authors included an important control, namely the knock-down of BACE2 (Fig. EV4E). However, they did not compare the changes to the ones observed in the pharmacological assay in Fig. 5B. This needs to be added (e.g. Venn diagram plus a brief description in the results section) to ensure that the pharmacological and genetic approach generally lead to similar changes and not only for the protein picked by the authors.

2. I would like to see the quantification data of figure 5B in an easily understandable manner. The authors refer to dataset EV9. However, what is needed is to add (as separate columns) the mean value and p value so that I do not need to compute it myself, but can instead immediately choose one data point from the volcano and find the corresponding values in the table and verify them. This table needs to be updated.

Minor points:

3. Figure EV3H: the drug concentration in the panel is given in microgram, but should be micromolar.

4. I had asked for a definition of "maturation", e.g. in the context of APP. The authors answered that they use "maturation" relating to "maturation into fibrils". For membrane proteins, "maturation" is typically used to refer to PTM acquisition (in particular complex glycosylation) along the secretory pathway. To enhance the clarity of the manuscript, the authors should be more specific, e.g. by writing the whole expression "maturation into fibrils".

Referee #2:

In reference to my previous points, the authors have not addressed enough my concerns to strengthen their claims. The manuscript remains very correlative, and in the end the experimental evidence does not support the proposed model.

point A: the observation of increased amyloid aggregates remains without any mechanism, apart from the overexpression of BACE2 in metastatic cells. I had proposed this as a possible manner to enrich the core message and to compensate weaknesses in other parts of the manuscript, but this was not expanded.

Actually, it remains unclear which is the mechanism that drives increased amyloid fibrils in primary vs metastatic cells. We have demonstrated that BACE2 regulates amyloid fibrils maturation specifically in metastatic cells, however we do not know why and how BACE2 is activated in metastatic but not in primary cells. In vivo studies on rat models highlighted that high glucose upregulates BACE expression and activity through HIF-1 α and LXRA regulated lipid raft reorganization, leading to A β production (Lee, H., Ryu, J., Jung, Y. et al. High glucose upregulates BACE1-mediated A β production through ROS-dependent HIF-1 α and LXRA/ABCA1-regulated lipid raft reorganization in SK-N-MC cells. Sci Rep 6, 36746 (2016)). We might hypothesize that BACE activation in melanoma could be metabolically regulated thus being a consequence of the Warburg effect; obviously this is extraordinary interesting but it has not been investigated in this study. As we observed the presence of amyloids fibrils specifically in metastatic vs primitive cells we focused our attention on what these fibrils might cause in metastasis and how we can inhibit their production, therefore our study is focused downstream amyloids production and not on their origin.

point B: the authors should be careful in writing (text and figures) "proliferation" when they just count the number of cells. The new data still do not provide any direct proof that the cell cycle is altered, for which a PI staining or EdU incorporation would easily provide support. Annexin V staining excludes death from classical apoptosis, but cell number could still change due to other types of death.

Moreover, please be consistent on the Y axis of graphs: in figure 6, for example, A, B, C, D are all MTT assays, but his is labeled "cell number %" in A-B and "cell viability, % of control" in C-D which may lead the reader to believe these are different and more specific assays. What is quantified is just (indirectly) the number of cells.

We thank the reviewer for highlighting this critical point. Indeed, Annexin V is an apoptotic marker that cannot exclude different type of cell death, however we performed a "cell viability" assay in IGRs cells after NB-360 (MUSE reagent described in materials and methods) gathering information on the number of both alive and dead cells. As shown in Fig EV3H and Fig below, cell viability is not affected upon NB360 treatment, demonstrating that BACE inhibition decreases proliferation without affecting the number of dead cells. We specified more clearly in the revised manuscript that we performed different and more specific assays to assess viability and cell number %.

We also apologize for the different labels in fig 6 A-D, we have corrected them.

point C: the data can support the notion that secreted PMEL fibrils activate YAP, provided that the authors can show that YAP localization in cells treated with NB360 is rescued by adding back IGR37 CM or PML fibrils.

We have performed a rescue experiment demonstrating that the administration of PMEL fibrils to NB360 treated cells is able to restore the expression of CTGF. This experiment also indicates that, despite the broad effect of the dual BACE inhibitor, PMEL fibrils alone have an effect on YAP transcriptional activity similarly to what happens when YAP is activated by canonical mechanotransduction signals. We have reported this experiment in Fig 5J in the revised submission.

Any reference to mechanotransduction is otherwise inappropriate. The sole correlation with FAK activity and Agrin secretion cannot be claimed to be "mechanotransduction" in absence of specific measures of the cytoskeleton, cell contractility etc., and without a clear demonstration for the FUNCTIONAL involvement of these players, starting from Agrin for which the data are only correlative. So, any mention to mechanotransduction should be avoided in the title, abstract and text. This can be of course discussed, in light of correlative evidence.

The same applies for Agrin: one thing is showing correlation with a secreted protein, another is providing experimental functional support that Agrin is REQUIRED downstream of fibrils, which is not provided here - and, thus, should not be claimed to be the mechanism by which YAP is being regulated.

Collectively, the link with YAP thus remains an observation without a mechanism, and without a functional relevance for the BACE/fibril phenotypes.

We are conscious that we did not measure fibrils physical properties that could support a direct role of fibrils in mediating mechanotransduction, therefore we only discussed this part based on known intrinsic rigidity of amyloids structures. However, we demonstrated that PMEL fibrils administration is able to activate known and well recognized markers of mechanotransduction. In particular, we showed that PMEL fibrils not only promote CTGF expression (Fig 5I) but increase also the expression of Agrin and the level of FAK phosphorylation (fig EV4 J-K) similarly to what is reported when cells are grown in a stiff matrix. Indeed, Lachowski et al. (FAK Controls the Mechanical Activation of YAP, a Transcriptional Regulator Required for Durotaxis FASEB J. 2018 Feb;32(2):1099-1107), demonstrated that an increase in FAK phosphorylation is necessary to induce YAP nuclear

translocation and activation. In conclusion, we cannot exclude that amyloid fibrils have also other effects but we clearly show that they impinge on the mechanotransduction pathway. Further work is necessary to clarify if this is a direct link or if and which other molecules are required.

We have changed the title and the discussion accordingly.

point D: the new data with the new BACE inhibitor are better in line with the proposed model. Still, at face value, all the rest of the functional evidence supporting the author's claims is based on the use of the least specific inhibitor, which may raise some specificity issues. Moreover, the functional role of the fibrils is almost uniquely based on the use of BACE inhibitors, which will also inhibit other pathways in cells beyond fibrils. So, to claim that the observed effects of BACE are through fibrils, the authors should provide key data throughout the paper (proliferation, YAP, chemotherapy etc) showing that an alternative means of inhibiting fibrils (e.g. knockdown of PMEL?) induces the same phenotypes.

We are pleased that the reviewer considers the new BACE inhibitor experiments better in line with the proposed model. However, he/she claims that the observed effects of BACE should be supported by alternative means of inhibiting fibrils.

Concerning YAP, we proved that amyloid fibrils are able to directly drive its transcriptional activation and rescue BACE inhibition (see answer to point C).

Concerning proliferation and chemotherapy, even though we do not provide further evidences, it has already been proved that PMEL KD is in line with our model (Effects of microRNA-136 on melanoma cell proliferation, apoptosis, and epithelial–mesenchymal transition by targeting PMEL through the Wnt signaling pathway Wang JJ et al Biosci Rep. 2017). Indeed, Wang JJ et al. reported that PMEL silencing affects mouse melanoma cells proliferation similarly to what we observed by BACE inhibition. Moreover, it has also been reported that PMEL expression is associated with resistance to chemotherapy in melanoma (Association of MITF and other melanosome-related proteins with chemoresistance in melanoma tumors and cell lines. Johansson, C. et al. Melanoma Research 2013). In particular, siRNA inhibition of PMEL in the MNT-1 melanoma cells sensitized the cells to both paclitaxel and cisplatin in line with the increased chemo-sensitivity induced by BACE inhibition described in our work.

Referee #3:

The authors have reasonably addressed my previous points, but four more points need to be addressed:

Major points:

1. The authors included an important control, namely the knock-down of BACE2 (Fig. EV4E). However, they did not compare the changes to the ones observed in the pharmacological assay in Fig. 5B. This needs to be added (e.g. Venn diagram plus a brief description in the results section) to ensure that the pharmacological and genetic approach generally lead to similar changes and not only for the protein picked by the authors.

We compared the changes observed in the pharmacological assay in Fig. 5B. with the ones observed upon knock-down of BACE2 in Fig. EV4E. We have added in fig EV4E a Venn diagram showing that more than 250 proteins were affected by both treatments, attesting that the pharmacological and the genetic approach share a similar behavior. As discussed in the manuscript, among the common proteins we confirmed that PMEL secretion was affected as well as CTGF expression.

2. I would like to see the quantification data of figure 5B in an easily understandable manner. The authors refer to dataset EV9. However, what is needed is to add (as separate columns) the mean value and p value so that I do not need to compute it myself, but can instead immediately choose one data point from the volcano and find the corresponding values in the table and verify them. This table needs to be updated.

Fold changes and p values of the proteins related to figure 5B are reported in dataset EV10. We are sorry for forgetting to properly refer it to the volcano plot. As requested, we also added the mean values of the z-score for each protein in the revised EV10 table.

Minor points:

3. Figure EV3H: the drug concentration in the panel is given in microgram, but should be micromolar.

Sorry for the mistake, we corrected microgram with micromolar in Figure EV3H.

4. I had asked for a definition of "maturation", e.g. in the context of APP. The authors answered that they use "maturation" relating to "maturation into fibrils". For membrane proteins, "maturation" is typically used to refer to PTM acquisition (in particular complex glycosylation) along the secretory pathway. To enhance the clarity of the manuscript, the authors should be more specific, e.g. by writing the whole expression "maturation into fibrils".

As suggested, we changed "maturation" into "maturation into fibrils"

Angela Bachi
IFOM-FIRC Inst. of Molecular Oncology
Via Adamello 16
Milano 20139
Italy

Dear Dr. Bachi,

Thank you for the submission of your final revised research manuscript to EMBO reports. I have now received the report from the referee that was asked to re-evaluate your study, which can be found at the end of this email. The referee now supports the publication of your study. As I think that also the concerns by referee #1 have been adequately addressed, I am very pleased to accept your manuscript for publication in the next available issue of EMBO reports. Thank you for your contribution to our journal.

At the end of this email I also include important information about how to proceed. Please ensure that you take the time to read the information and complete and return the necessary forms to allow us to publish your manuscript as quickly as possible.

As part of the EMBO publication's Transparent Editorial Process, EMBO reports publishes online a Review Process File to accompany accepted manuscripts. As you are aware, this File will be published in conjunction with your paper and will include the referee reports, your point-by-point response and all pertinent correspondence relating to the manuscript.

If you do NOT want this File to be published, please inform the editorial office within 2 days, if you have not done so already, otherwise the File will be published by default [contact: emboreports@embo.org]. If you do opt out, the Review Process File link will point to the following statement: "No Review Process File is available with this article, as the authors have chosen not to make the review process public in this case."

Should you be planning a Press Release on your article, please get in contact with emboreports@wiley.com as early as possible, in order to coordinate publication and release dates.

Thank you again for your contribution to EMBO reports and congratulations on a successful publication. Please consider us again in the future for your most exciting work.

Yours sincerely,

Achim Breiling
Editor
EMBO Reports

Referee #3:

The authors have reasonably well addressed my previous points.

THINGS TO DO NOW:

You will receive proofs by e-mail approximately 2-3 weeks after all relevant files have been sent to our Production Office; you should return your corrections within 2 days of receiving the proofs.

Please inform us if there is likely to be any difficulty in reaching you at the above address at that time. Failure to meet our deadlines may result in a delay of publication, or publication without your corrections.

All further communications concerning your paper should quote reference number EMBOR-2020-50446V3 and be addressed to emboreports@wiley.com.

Should you be planning a Press Release on your article, please get in contact with emboreports@wiley.com as early as possible, in order to coordinate publication and release dates.

Corresponding Author Name: Angela Bachl

EMBOR-2020-50446V3